# Capacity Analysis of Vector Symbolic Architectures

## Abstract

Hyperdimensional computing (HDC) is a biologically-inspired framework which represents symbols with high-dimensional vectors, and uses vector operations to manipulate them. The ensemble of a particular vector space and a prescribed set of vector operations (e.g., addition-like for "bundling" and outer-product-like for "binding") form a *vector symbolic architecture* (VSA). While VSAs have been employed in numerous learning applications and have been studied empirically, many theoretical questions about VSAs remain open. In this paper, we analyze the *representation capacities* of four common VSAs: MAP-I, MAP-B, and two VSAs based on sparse binary vectors. "Representation capacity" here refers to bounds on the dimensions of the VSA vectors required to perform certain symbolic tasks, such as testing for set membership and estimating set intersection sizes for two sets of symbols, to a given degree of accuracy. We also analyze the ability of a novel variant of a Hopfield network (a simple model of associative memory) to perform some of the same tasks that are typically asked of VSAs. In addition to providing new bounds on VSA capacities, our analyses establish and leverage connections between VSAs, "sketching" (dimensionality reduction) algorithms, and Bloom filters.

## 1 Introduction

*Hyperdimensional computing* (HDC) is a biologically inspired framework for representing symbolic information (Kanerva, 2009; Kleyko et al., 2021a). In this framework, we represent different symbols using high-dimensional vectors (hypervectors) called *atomic vectors*, sampled from a natural vector distribution, such as random signed vectors or sparse binary vectors. We then use standard arithmetic or bit-wise operations on these vectors to perform symbolic operations like associating or grouping symbols. The distribution over atomic vectors, together with their corresponding operations, form a *vector-symbolic architecture* (VSA) (Schlegel et al., 2020).

Many features of VSAs are inspired by aspects of the human brain and memory. They are robust to noise, and the fact that all entries of the vectors are symmetric (i.e. don't correspond to specific features) allows VSA computations to be highly parallelizable. Most VSA vector entries are $\{0, \pm 1\}$, so there is no need for floating point arithmetic, and they admit energy-efficient and low-latency hardware implementations (Kleyko et al., 2022). The atomic vectors are typically chosen randomly, with a distribution such that they are likely to be pairwise *near-orthogonal*. While we encounter "the curse of dimensionality" in many algorithmic and machine learning problems, the ability to fit many near-orthogonal vectors into $d$-dimensional space is a kind of "blessing of dimensionality."

While VSAs sometimes do not achieve the same performance as neural networks do on certain classification tasks, VSA-based classifiers require much less time and space to train and store Ge & Parhi (2020). There is also much research at the intersection of VSAs and deep learning; for instance, VSA vector representations could be good neural network inputs, and since all vector representations have the same size, this could be a better way to support inputs with different numbers of features (Kleyko et al., 2019a; Alonso et al., 2021). In addition, VSAs have found numerous applications outside of learning, including the modeling of sensory-motor systems in smaller organisms (Kleyko et al., 2015a;b), combining word embeddings to form context embeddings (Kanerva et al., 2000; Sahlgren, 2005; Jones & Mewhort, 2007), and the processing of heart rate respiratory data and other biological times series data (Kleyko et al., 2018a; 2019b; Burrello et al., 2019). They are

Table 1: A summary of MAP-I, MAP-B, and Binary Sparse VSAs.

| VSA | Atomic Vector | Bundling | Binding |
|---|---|---|---|
| MAP-I | $\{\pm 1\}^m$ | Elem.-wise addition | Elem.-wise mult. |
| MAP-B | $\{\pm 1\}^m$ | Elem.-wise add., round to $\pm 1$ | Elem.-wise mult |
| Bloom filters | $k$-sparse, $\{0,1\}^m$ | Elem.-wise addition, cap at 1 | (Out of scope) |
| Counting Bloom filters | $k$-sparse, $\{0,1\}^m$ | Elem.-wise addition | (Out of scope) |

also promising tools for bridging the gap between symbolic data, such as word relationships, and numerical data, such as trained word embeddings (Quiroz-Mercado et al., 2020).

In addition to exploring the performance of VSAs in such applications, there have also been a few works that study the *representation capacity* of many VSAs (Neubert et al., 2017; Mirus et al., 2020). The representation capacity refers to lower bounds on the VSA dimension, needed so that we can reliably perform certain symbolic tasks, such as computing set intersection sizes $|X \cap Y|$ for two sets of symbols $X$ and $Y$ (with possibly different cardinalities). These lower bounds then translate into a design choice (minimum required dimension) for VSA applications. Here we understand reliability in the setting of generating the atomic vectors at random, and asking for good-enough accuracy with high-enough probability. Similarly, another way to quantify representation capacity is: if we tolerate a failure probability of $\delta$, can we bound the number of sets/objects we can encode?

**Overview and significance of our contributions:** The goal of this paper is to provide theoretical analyses of the representation capacities of four VSA models for different symbolic operations. In general, there are more models employed in practice; see (Schlegel et al., 2022; Kleyko et al., 2021b) for longer catalog of VSAs. We also assemble a set of tools and frameworks for any future computations and high-probability bounds on VSA capacity. We focus on four popular VSA families, plus a novel variant of Hopfield networks. Table 1 shows the atomic vector initializations of the four, as well as their bundling and binding operators. As surveyed by Schlegel et al. (2022), "MAP" stands for "Multiply-Add-Permute," and "I" and "B" reference the values that non-atomic vectors take ("integer" and "binary," respectively). While several binding operators have been proposed for Binary Sparse VSAs, the study of these operators is outside the scope of this paper. A key aspect of our analyses is that we translate several VSA operations of interest into the language of sketching matrices (used for dimensionality reduction) or of hashing-based data structures (like Bloom Filters). These connections have not been made explicit, or leveraged previously in VSA analysis.

In several of our analyses, we rely on the Johnson-Lindenstrauss (JL) property, which is foundational to algorithmic matrix sketching, the study of techniques for compressing matrices while approximately preserving key properties, such as least-squares solutions, see (Woodruff, 2014) for details. From the perspective of sketching, our results on permutations, bundles of $k$-bindings, and Hopfield networks can be regarded as extensions of sketching theory using structured matrices with less randomness than the most direct approaches. The bundling operator for MAP-B is per-coordinate the *Majority function* in the setting of the analysis of Boolean functions, and is a simple *linear threshold function*. The analysis of the membership test for MAP-B that we give is close to that for analyzing the *influence* of variables for the majority function (cf. (O'Donnell, 2014, Exer. 2.2)). Our analysis of MAP-B extends to other operations, including a bundling of bindings, which are related to *polynomial threshold functions* of degree $k$, where $k$ depends on the maximum number of symbols that are bound together. Two of the VSAs we analyze are based on Bloom filters, which are space-efficient data structures for testing set membership. They have seen extensive study since their introduction fifty years ago by Bloom (1970), but we have not found rigorous analyses of their performance in the setting of set intersection. Our results are novel contributions in this regard.

**Related works:** Some recent works in the VSA literature that have inspired our own work also study the capacity problem, but they analyze more restricted VSA systems and/or use different analysis approaches. To our knowledge, we are the first to formally analyze VSAs in terms of sketching, using tools like the JL property, though Thomas et al. (2022) does mention the sparse JL transform, but under different context. Thomas et al. (2021b) provides a theoretical analysis of VSAs in terms of a measure called "incoherence," which quantifies the size of the "cross-talk" (dot product) between two independently initialized VSA vectors. They prove results on membership testing and set intersection of bundles, membership testing in key-value pairs, and show pairwise near-orthogonality of vectors and their coordinate permutations in terms of incoherence. Their work is in the linear setting (which pertains to MAP-I), and extends, as does ours via known results,

| VSA | Op | Expression | Dimension $m$ | Thm. or Section |
|---|---|---|---|---|
| MAP-I | S | $\|\bar{S}v\|^2$ | $O(\varepsilon^{-2}\log(1/\delta))$ | Lem 2 |
| Sparse JL | S | $\|P_{\text{SJL}}v\|^2$ | $O(\varepsilon^{-2}\log(1/\delta))$ | §B.1.1 |
| SRHT | S | $\|P_{\text{SRHT}}v\|^2$ | $O(\varepsilon^{-2}\log(1/\delta)\log^4 d)$ | §B.1.1 |
| MAP-I | SS | $\|\bar{S}_{R,L}v\|^2$ | $O(\varepsilon^{-2}L^2\log(L/\delta))$ | Thm 8 |
| MAP-I | SS | $\|\bar{S}_{R,L}v\|^2$ | $O\left(\varepsilon^{-2}K^2\log(K/(\varepsilon\delta))\right)$ | Thm 9 |
| MAP-I | BB | $\|\bar{S}^{\odot 2}v\|^2$ | $O(\varepsilon^{-2}\log(\|v\|_1/\varepsilon\delta)^3)$ | Thm 11 |
| MAP-I | BB | $\|\bar{S}^{\odot k}v\|^2$ | $O(\varepsilon^{-2}C^{k\log k}\log^{k+1}(k\|v\|_1/(\varepsilon\delta)))$ | Cor 12 |
| Hopfield± | S | $\text{tr}(\bar{S}VD\bar{S}^\top)$ | $O(\varepsilon^{-1}\log(d/\delta)^2)$ (space is $m^2$) | Thm 14 |
| MAP-B | MS | $\text{sign}(Sv)^\top S_{*j} \geq \tau_b$ | $O(\|v\|_1\log(d/\delta))$ | Thm 15 |
| MAP-B | MSS | $\text{sign}(S_{R,L}v)^\top R^{\lfloor j/d\rfloor}S_{*,j\%d} \geq \tau_b$ | $O(\|v\|_1 L\log(Ld/\delta))$ | Thm. 38 |
| MAP-B | MKV | $\text{sign}(S^{\odot 2}v)^\top S_{*j}^{\odot 2} \geq \sqrt{2}\tau_b$ | $O(\|v\|_1\log(d/\delta))$ | Thm 40 |
| Bloom | SI | $h_{m,k}(Bv \cdot Bw)$ | $O(k\varepsilon^{-1}(n_v n_w + n^2 + \varepsilon(n+n_w)))$ $k = O(\varepsilon^{-1}\log(1/\delta))$ | Thm 18 |
| Counting Bloom | GSI | $Bv \curlywedge Bw$ | $O(k\varepsilon^{-1}n_v n_w)$ $k = O(K_b\varepsilon^{-1}\log(1/\delta))$ | Thm 19 |

Table 2: We compile the bounds in this paper for various architectures and operations. Here $\delta$ is a bound on the failure probability; $m$ is the VSA dimension; $d$ is the size of the groundset of symbols; $S, \bar{S}$ are defined in Def. 1; $L$ is the sequence length; $\bar{S}_{R,L}$ denotes a sequence, Def. 7; $\bar{S}^{\odot k}$ denotes a bundle of bindings, Def. 10; and $v \in \{0,1\}^d$ for sets of singletons, $v \in \{0,1\}^{Ld}$ for sequences, $v \in \{0,1\}^{\binom{d}{k}}$ for bundles of $k$ bindings.

The **first block of rows** are bounds for norm-preserving *linear* operators, so that relative error $\varepsilon$ bound for each in estimating $\|v\|$ translates to an error bound for estimating $\|u-v\|$ for many pairs of vectors $u, v$, as in Cor. 3; dot products, as in Cor. 4; and in computing set intersection sizes under a variety of conditions on the set sizes, Thm. 5.

**First block notation:** operation $S$ is estimation of size of a set; *SS* is size of sequence of sets; *BB* is size of a Bundle of Bindings (set of bound key-value pairs); matrices $P_{\text{SJL}}, P_{\text{SRHT}}$ are sparse JL matrix and SRHT matrix, as discussed in §B.1.1; $K$ the maximum number of times a symbol appears in a sequence; $V$ is a diagonal matrix version of $v$; $D$ is a diagonal sign matrix.

The **second block** are operations with non-linear steps. $\varepsilon$ is the *additive* estimation error.

**Second block notation:** operation *MS* is set membership test; *MSS* is membership in a sequence of sets; *MKV* is membership in a bundle of bindings, where the bindings are key-value pairs; *GSI* is generalized set intersection (estimation of $v \curlywedge w = \sum_i \min\{v_i, w_i\}$); matrix $B$ is a sparse binary matrix, Def. 17; $\tau_b = \sqrt{2m\log(2d/\delta)}$; $k$ is the number of nonzeros per column of $B$; $h_{m,k}(z) = \frac{1}{k}\log_{1-1/m}(1-z/m)$, defined in Lem. 41; $v, w \in \{0,1\}^d$ for Bloom filters; $v, w \in \mathbb{Z}_{\geq 0}^d$ for Counting Bloom filters; $K_b = \|v-w\|_\infty \leq \max\{\|v\|_\infty, \|w\|_\infty\}$; finally, $n = v \curlywedge w$, $n_v = \|v\|_1 - n$, and $n_w = \|w\|_1 - n$, so $n_v + n_w = \|v-w\|_1$.

from sign matrices to matrices of independent sub-Gaussians. We add new bounds that include set intersections for bundles of $k$-bindings, and sequences of sets encoded with rotations (cyclic permutations). Our hypergraph framework for binding in §B.3 also allows us to extend our analysis for bundlings of bindings beyond only key-value pairs. Another related work is that of Kleyko et al. (2018b), which studies bundling in binary VSA systems, where atomic vectors are controlled by a sparsity parameter. Their model is slightly different from the Bloom-filter inspired binary VSAs that we analyze, and their bundling operator includes a "thinning" step, where the support of the bundle gets reduced to the size of a typical atomic vector. Their analysis of VSA capacity uses Gaussian approximations holding in the limit, and some heuristics to get around the thinning step. We formally analyze the capacity of bundling without thinning, using concentration inequalities. Kleyko et al. (2018b) also conduct several empirical analyses and simulations of sparse binary VSAs, which may be of independent interest.

**Summary of contributions:** A summary of most of our results are given in Table 2. A few results here *not* in the table: analyses of Hopfield nets using standard concentration results (§C.1); and of the decay in "signal" in MAP-B when bundling operation trees have nontrivial depth (§D.1.2).

## 2 TECHNICAL OVERVIEW

We begin with background details on VSAs, and an overview of the techniques used in this paper.

**Notation and Terminology:** We summarize some notation in the following list.

- $\lfloor a \rceil$ is the nearest integer to $a \in \mathbb{R}$;
- $\text{sign}(a)$ for $a \in \mathbb{R}$ is 1 if $a > 0$, $-1$ if $a < 0$ and $\pm 1$ with equal probability if $a = 0$;
- $\text{sign}_{\geq}(a)$ for $a \in \mathbb{R}$ is 1 if $a \geq 0$, and $-1$ otherwise
- For $a, b \in \mathbb{R}$, $a = b \pm \varepsilon$ means that $|a - b| \leq \varepsilon$, so that $a = b(1 \pm \varepsilon)$ means that $|a - b| \leq b\varepsilon$;
- $v \circ w$ denotes the Hadamard (elementwise) product of two vectors, $(v \circ w)_i = v_i w_i$;
- $v \wedge w$ denotes the elementwise minimum of two vectors $v$ and $w$, $(v \wedge w)_i = \min\{v_i, w_i\}$; for scalar $a$ and vector $v$, $a \wedge v$ is the vector with $(a \wedge v)_i = \min\{a, v_i\}$;
- $v \barwedge w$ denotes $\sum_i \min\{v_i, w_i\}$;
- $\text{supp}(v) = \{i \in [d] \mid v_i \neq 0\}$, for $v \in \mathbb{R}^d$. For diagonal $V \in \mathbb{R}^{d \times d}$, let $\text{supp}(V) = \{i \in [d] \mid V_{ii} \neq 0\}$;

**Background on VSAs:** To define a VSA, we need to first choose a distribution for sampling the atomic vectors. For every individual symbol we want to introduce (excluding associations between multiple symbols), we will independently sample a random $m$-dimensional vector from a distribution, such as the uniform one over $\{\pm 1\}^m$ (i.e. sign vectors), or a random $k$-sparse binary ($\{0, 1\}^m$) vector. VSAs are equipped with the following operations; permutation is not always required.

- **Similarity Measurement** ($\langle \cdot, \cdot \rangle$)**:** In almost all VSAs in this paper, we use cosine similarity (i.e. dot product $\langle u, v \rangle$) to measure the similarity between the symbols that $u$ and $v$ represent.
- **Bundling** ($\oplus$)**:** We use bundling to represent a "union" of several symbols. The goal is for $u \oplus v$ to be similar to both $u$ and $v$.
- **Binding** ($\otimes$)**:** Binding is often used to associate symbols as (key, value) pairs. Here, we want $u \otimes v$ to be nearly orthogonal to both $u$ and $v$.
- **Permutation** ($\pi$)**:** If $\pi$ is a permutation, let $\pi(v)$ be the vector whose $i$-th entry is $v_{\pi^{-1}(i)}$. We use $[v_1; \ \pi(v_2); \ \pi^2(v_3); \ \ldots; \pi^{L-1}(v_L)]$ to encode an *ordered sequence* of symbols.
- **Cleanup** [1]**:** Let vector $x$ represent a set $X \subseteq \mathcal{X}$, where $\mathcal{X}$ is the universe of all symbols. Given a dictionary of symbols $S \subseteq \mathcal{X}$, we want to compute the elements in $X \cap S$.

**Example 1.** *We can represent categorical data (Agresti, 2012) using the bundle of bindings:*

$$\vec{y} = \bigoplus_{i \in \text{ features}} \vec{f_i} \otimes \vec{v_i}$$

*Here, $\vec{f_i}$ is a VSA vector encoding the concept "feature i," while $\vec{v_i}$ is a VSA vector encoding the value of feature $i$. To train a classifier on the VSA representations $\{\vec{y}\}$, we can average the training vectors over each class to get an exemplar for the class.*

$$\vec{c_i} = \frac{1}{\#\{z : \ label(z) = i\}} \sum_{z: \ label(z) = i} z$$

*To classify a new input, we first compute its VSA encoding. We then return the class whose representative vector $c_i$ is closest to it (using similarity measurement).*

We will study a few basic predicates on VSAs; one is *membership testing*: given a vector $x$ representing a bundle of atomic vectors, and an atomic vector $y$, determine if $y$ is in the set represented by $x$. Another predicate is *set intersection*: if $y$ also represents a set, estimate the size of the intersection of the sets represented by $x$ and $y$. Here these sets might be sets of vectors bound together, or permuted. Estimation of intersection size is a natural extension of membership testing, and is helpful in applications where VSA represent, for example, sets of properties, where intersection and difference size are rough measure of relatedness.

**Bundling as the output of a linear map.** Our central approach is to cast VSA vectors as the outputs of linear transformations, followed by nonlinear maps for some of the VSAs. Via the linear maps,

---

[1]Cleanup will not be a focus of our paper. If we have access to the individual vector representations of elements in $S$, we can execute cleanup as a set of (parallel) membership tests, which use similarity measurement.

we can translate between a very simple vector representation for symbols (described below) and the more robust, fault-tolerant, and (in some settings) lower-dimensional ones used in VSAs.

A simple vector representation of elements in a universe $\mathcal{X}$ is their one-hot encoding as unit binary vectors $e_i \in \{0, 1\}^d$, where $d \equiv |\mathcal{X}|$. Then, set union corresponds to adding vectors: a set $X \subset \mathcal{X}$ is a characteristic vector $v \in \{0, 1\}^d$ with $v = \sum_{i \in X} e_i$. The size of $|X|$ corresponds to the squared Euclidean norm $\|v\|^2$. For two sets $X, Y \subseteq \mathcal{X}$ with corresponding characteristic vectors $v$ and $w$, respectively, $|X \cap Y|$ is simply $v^\top w$; similarly, their symmetric difference $|X \Delta Y|$ is $\|v - w\|^2$. Thus an embedding of these one-hot encodings, linear or non-linear, that approximately preserves distances and/or dot products of vectors (which is particularly challenging for binary vectors), can be used to maintain sets and the sizes of their intersections and symmetric differences. With this in mind, we can regard the atomic vectors of a VSA as the columns of a random matrix $P$ (so the embedding of vector $x$ is $Px$, possibly with a non-linearity applied), and for appropriate random $P$, we will obtain such embeddings. From this perspective, VSAs perform dimensionality reduction from characteristic vectors in $\{0, 1\}^d$ to an $m$-dimensional space, for $m \ll d$.

**Overview of Tools and Techniques:** Our analysis of MAP-I establishes the JL property for a variety of random sign matrices, some with dependent entries, that are used to encode bundles, sequences, and bundles of bindings. In the case of sequences and bindings, the entries of the sign matrices we analyze will not be independent, which complicates our analysis; we get around this using standard concentration results like McDiarmid's inequality (Theorem 21) and a version for when the bounded differences hold with high probability (Corollary 23). Our analysis of MAP-B also uses these concentration techniques and also borrows some ideas from Boolean Fourier analysis. For Hopfield nets and our variation Hopfield±, we also use standard concentration results like the Hanson-Wright inequality. Bernstein-like variation of McDiarmid, Theorem 24, is key to our analysis of Bloom filters.

## 3 STATEMENTS OF RESULTS

In the remainder of the paper, we will present concrete statements of our results, and highlight some of the proof ideas, with an emphasis again on connections to sketching and data structures. Each subsection here (roughly organized by each different VSA we analyze) will have a full section in the appendix where we give their proofs.

### 3.1 ANALYSIS OF MAP-I USING JOHNSON-LINDENSTRAUSS

**Bundling and Set Intersection.** For MAP-I, we choose $P$ as a scaled sign matrix $\bar{S}$.

**Definition 1.** *A* sign vector *$y \in \{\pm 1\}^m$ (also called a Rademacher vector) is a vector with independent entries, each chosen uniformly from $\{\pm 1\}$. A* sign matrix *$S \in \{\pm 1\}^{m \times d}$ has columns that are independent sign vectors, and a* scaled *sign matrix $\bar{S} = \frac{1}{\sqrt{m}} S$ where $S$ is sign matrix.*

Thus, $Se_i$ is a random sign vector. For MAP-I, the bundling operator is addition, so a set $X$ can be represented as $\bar{S}v$, with $v = \sum_{i \in X} e_i$. (More typically, MAP-I would use the unscaled sign matrix $S$, but $\bar{S}$ is convenient for analysis and discussion.) It is known that with high probability, for $m$ sufficiently large, that $\bar{S}$ satisfies the *Johnson-Lindenstrauss (JL) property*, which is the norm-preserving condition in the lemma below.

**Lemma 2** (Johnson (1984); Achlioptas (2003), JL). *Suppose $\bar{S} \in \frac{1}{\sqrt{m}}\{-1, 1\}^{m \times d}$ is a scaled sign matrix (described in Def. 1). Then for given $\delta, \epsilon > 0$ there is $m = O(\epsilon^{-2} \log(1/\delta))$ such that for given vector $v \in \mathbb{R}^d$, it holds that $\|\bar{S}v\| = \|v\|(1 \pm \epsilon)$, with failure probability at most $\delta$.*

This immediately tells us the dimension $m$ required to estimate set sizes up to a multiplicative factor of $\varepsilon$, with failure probability $\delta$. For $X, Y \subseteq \mathcal{X}$, let $v$ and $w$ denote their characteristic vectors. An immediate consequence of the JL lemma is that the symmetric difference size $|X \Delta Y| = \|v - w\|^2$ can be estimated with small relative error as well.

**Corollary 3.** *If random matrix $P$ satisfies the JL property (Lemma 2), then for given $\delta, \epsilon > 0$, and a set $\mathcal{V} \subset \mathbb{R}^d$ of cardinality $n$, there is $m = O(\epsilon^{-2} \log(n/\delta))$ such that with failure probability $\delta$, for all pairs $v, w \in \mathcal{V}$, $\|P(v - w)\|^2 \in \|v - w\|^2(1 \pm \epsilon)$.*

The JL lemma also implies that $(Pv)^\top (Pw) = v^\top P^\top P w$ concentrates around $|X \cap Y| = v^\top w$.

**Corollary 4.** *Suppose the random matrix $P$ satisfies the JL property (Lemma 2). Then for $v, w \in \mathbb{R}^d$, there is $m = O(\epsilon^{-2} \log(1/\delta))$ so that $v^\top P^\top P w = v^\top w \pm \epsilon \|v\| \|w\|$ with failure probability $\delta$.*

Estimation of the cosine of the angle between $v$ and $w$, which is $|X \cap Y|/\sqrt{|X| \cdot |Y|}$, up to additive error $\epsilon$, is also now immediate. The goal of our MAP-I bundling section is to use the above lemmas and corollaries to prove the following theorem.

**Theorem 5.** *Suppose random matrix $P$ satisfies the JL property (Lemma 2). Given $M$ pairs of characteristic vectors $v, w \in \{0, 1\}^d$ such that for every pair $v, w$, $\|v\|_1 \|w\|_1 \leq N$, then there is $m = O(N \log(M/\delta))$ such that $\lfloor v^\top P^\top P w \rceil = v^\top w$ for all $M$ pairs with probability $\geq 1 - \delta$.*

In short, we inherit much of our capacity analysis of bundling in MAP-I through known results about the JL property.

**Rotations via JL property.** One operation used in VSAs to expand on the number of nearly orthogonal vectors available for use is via permutations of the entries; this is the "P" in the **MAP** VSA systems introduced by Gayler (1998). These can be encoded in permutation matrices $R \in \{0, 1\}^{m \times m}$. We will focus on cyclic permutations.

**Definition 6.** *Let $R \in \mathbb{R}^{m \times m}$ denote the permutation matrix implementing a rotation, so that $R_{m,1} = 1$ and for $i \in [m-1]$, $R_{i,i+1} = 1$, with all other entries equal to zero.*

For a random sign vector $y$, and such a rotation matrix $R$ (or indeed for any permutation matrix with few fixed points), $Ry$ is nearly orthogonal to $y$, and the vectors $R^\ell y$ for $\ell = 0, 1 \ldots L$ are pairwise mutually orthogonal with high probability, if $L$ is not too large. However, the entries of these vectors are not independent, so additional care is needed in analyzing them.

Permutations (and specifically, rotations) can be used to encode a *sequence* of atomic vectors $x^\ell$ as a sum $\sum_\ell R^\ell x^\ell$. We can also encode a *sequence of sets* with characteristic vectors $v_{(\ell)}$.

**Definition 7.** *For $R \in \mathbb{R}^{m \times m}$, $S \in \mathbb{R}^{m \times d}$ and integer $L \geq 0$, let $S_{R,L} \in \mathbb{R}^{m \times Ld}$ denote*

$$[S \; RS \; R^2 S \; \ldots \; R^{L-1}S].$$

*For a sequence of $v_{(0)}, v_{(1)}, \ldots v_{(L-1)} \in \mathbb{R}^d$, let $v \in \mathbb{R}^{Ld}$ denote $v \equiv [v_{(0)} \; v_{(1)} \; \ldots \; v_{(L-1)}]$.*

The sequence given by $v$ in the definition can be represented in MAP-I as a single vector $\bar{S}_{R,L} v$. We first find $m$ such that $\bar{S}_{R,L}$ satisfies the JL property for general $v \in \mathbb{R}^{Ld}$.

**Theorem 8.** *Given scaled sign matrix $\bar{S} \in \mathbb{R}^{m \times d}$, rotation matrix $R \in \mathbb{R}^{m \times m}$ as in Def. 6, integer $L > 0$, and $\bar{S}_{R,L}$ as in Def. 7. Then for $v \in \mathbb{R}^{Ld}$ as defined above,*

$$|\|\bar{S}_{R,L} v\|^2 - \|v\|^2| \leq 3\epsilon \|v\|_{L,1}^2 \leq 3L\epsilon \|v\|^2,$$

*with failure probability $6L^2 \delta$. It follows that there is $m = O((L/\varepsilon)^2 \log(L/\delta))$ such that with failure probability at most $\delta$, $\|\bar{S}_{R,L} v\|^2 = (1 \pm \varepsilon) \|v\|^2$.*

The proof of Theorem 8 relies on partitioning the rows of $\bar{S}$ so that within the rows of a single partition subset, the corresponding rows of $\bar{S}$ and $R^i \bar{S}$ can be treated independently.

We get tighter bound on $m$ when the $v_{(i)}$ are characteristic vectors: our bound on $m$ depends on $K$, the maximum number of times that a given symbol appears in $v$.

**Theorem 9.** *Given scaled sign matrix $\bar{S} \in \mathbb{R}^{m \times d}$, rotation matrix $R \in \mathbb{R}^{m \times m}$ (Def. 6), integer $L > 0$, and $\bar{S}_{R,L}$ as in Def. 7. For a sequence of vectors $v_{(0)}, v_{(1)}, \ldots v_{(L-1)} \in \{0, 1\}^d$, let $K \equiv \|\sum_{0 \leq j < L} v_{(j)}\|_\infty$. There is $m = O\left(K^2 \varepsilon^{-2} \log(K/(\varepsilon\delta))\right)$ such that with failure probability $\delta$,*

$$|\|\bar{S}_{R,L} v\|^2 - \|v\|^2| \leq \epsilon \|v\|^2$$

We handle dependences between rows of $S$ using a version of McDiarmid's inequality (Corollary 23). As noted by Corollary 3 and Corollary 4, satisfying the JL property allows us to find $m$ so we can perform set intersection (dot product) and symmetric difference (subtraction) operations on vectors of form $[v_{(0)} \; v_{(1)} \ldots v_{(L-1)}]$ up to a multiplicative factor of $\varepsilon$ and failure probability $\delta$.

**Bundles of bindings: the JL property.** In MAP-I, we implement binding using the Hadamard (element-wise) products of atomic vectors. In this setting, we can compactly represent a bundling of $k$-wise atomic bindings using the matrix $\bar{S}^{\odot k}$.

**Definition 10.** *For sign matrix $S$, let $S^{\odot k} \in \frac{1}{\sqrt{m}} \mathbb{R}^{m \times \binom{d}{k}}$, where each column of $S^{\odot k}$ is the Hadamard (element-wise) product of $k$ different columns of $S$. The scaled version $\bar{S}^{\odot k}$ is $\frac{1}{\sqrt{m}} S^{\odot k}$. We clarify that $P^{\odot k \top}$ denotes $(P^{\odot k})^\top$.*

With this notation, a bundling of $k$-wise atomic bindings is $\bar{S}^{\odot k} v$, for $v \in \{0,1\}^{\binom{d}{k}}$. Note that $\mathbb{E}[\bar{S}^{\odot k} \bar{S}^{\odot k \top}] = \frac{\binom{d}{k}}{m} I_m$, while $\mathbb{E}[\bar{S}^{\odot k \top} \bar{S}^{\odot k}] = I_{\binom{d}{k}}$. In order to reason about intersections over bundles of $k$-bindings, or about the symmetric difference between the bundles of $k$-bindings, we again establish the JL property, this time for $\bar{S}^{\odot k} v$. We first present the result for the $k = 2$ case, which captures bundles of key-value bindings.

**Theorem 11.** *For $v \in \{0,1\}^{\binom{d}{2}}$, there is $m = O\left(\varepsilon^{-2} \log^3(\|v\|_1 / \varepsilon \delta)\right)$ so that for $\bar{S}^{\odot 2} \in \{\pm \frac{1}{\sqrt{m}}\}^{\binom{d}{2}}$, we have $\Pr[|\|\bar{S}^{\odot 2} v\|^2 - \|v\|^2| > \varepsilon \|v\|^2] \leq \delta$.*

We have a generalized result for all $k$ as well.

**Corollary 12.** *For scaled sign $\bar{S}^{\odot k} \in \{\pm \frac{1}{\sqrt{m}}\}^{\binom{d}{k}}$, $v \in \{0,1\}^{\binom{d}{k}}$, and $i \in [m]$, there is $C > 0$ and $m = O(\varepsilon^{-2} C^{k \log k} \log^{k+1}(k \|v\|_1 / (\varepsilon \delta)))$ such that $\Pr[|\|\bar{S}^{\odot k} v\|^2 - \|v\|^2| > \varepsilon \|v\|^2] \leq \delta$.*

The key challenge of establishing the JL property for bundles of bindings is the fact that the columns of $\bar{S}^{\odot k}$ are not independent; for instance, when $k = 2$, the columns corresponding to $i \otimes j$, $j \otimes k$, and $i \otimes k$ are dependent. To track these dependencies, we can represent $\bar{S}^{\odot k} v$ as a hypergraph when $v \in \{0,1\}^{\binom{d}{k}}$. Then, if we apply McDiarmid's inequality to the bundling of bindings, we can obtain the constants used in the bounded differences in terms of the degrees in the hypergraph.

**Roadmap for Proofs** In §B.1, we prove Theorem 5 and discuss how the JL frame extends to the sparse JL and Subsampled Randomized Hadamard Transforms (SRHTs). In §B.2, we prove Theorems 8 and 9. Finally, in §B.3, we prove Theorem 11 and Corollary 12.

## 3.2 Hopfield Nets and Hopfield±

In the early seventies, a simple model of associative memory, based on autocorrelation, was introduced in Nakano (1972); Kohonen (1972); Anderson (1972). In a paper that might be regarded as ushering in the "second age of neural networks," Hopfield (1982) reformulated that model, and described a dynamic process for memory retrieval. We analyze the *capacity* of Hopfield's model, where a collection of vectors constituting the memories is stored. (A survey of autoasssociative memories is given in Gritsenko et al. (2017).) The capacity problem is to determine the maximum number of such vectors a network can effectively represent in a model described below, and it is assumed that the vectors to be stored are sign vectors. In our setting, the memories are simply columns of a sign matrix $S \in \{\pm 1\}^{m \times n}$.

Much of our description of the model paraphrases that of Nakano (1972); Kohonen (1972); Anderson (1972). The neural network in Hopfield's model comprises a recurrent collection of neurons with pairwise synaptic connections, with an associated weight matrix $W \in \mathbb{R}^{m \times m}$, such that given input $x \in \{\pm 1\}^m$, a dynamic process ensues with vectors $x[\ell]$, where $x[0] = x$, and $x[\ell + 1] = \text{sign}_{\geq}(Wx[\ell])$. Here, $\text{sign}_{\geq}(z) = 1$ if $z \geq 0$, and $-1$ otherwise. When we reach a fixed point, i.e., $x[\ell + 1] = x[\ell]$, the process stops, and $x[\ell]$ is output. If a vector $x$ is a fixed point of the network, the network is said to "represent" $x$. Typically, the goal is to not only show the fixed-point property for some $x$, but also show that for $y \in \{0, \pm 1\}^m$ close to $x$ (i.e. $y^\top x$ is large), the network output is $x$ given input $y$, with high probability. The capacity problem is to determine how many vectors are represented by a network, as a function of $m$ and a given lower bound on $y^\top x$.

The weight matrix $W$ in a Hopfield network is $SS^\top - nI_m$, the sum of the outer products of the input vectors with themselves, with the diagonal set to zero. One motivation for this setup is that such a weight matrix can be learned by an appropriately connected neural network via a Hebbian learning process (Kosko, 1986). In Theorem 13 below, we give a bound on $m$ and $y^\top S_{*j}$ so that $\text{sign}_{\geq}((SS^\top - nI)y) = S_{*j}$ with bounded failure probability.

**Theorem 13.** *Given matrix $S \in \{\pm 1\}^{m \times n}$ with uniform independent entries, $j \in [n]$, and $\delta \in (0, 1]$. If $y \in \{0, \pm 1\}^m$ with $y^\top S_{*j} / \|y\| \geq 2\sqrt{n \log(2m/\delta)}$, then with failure probability at most $\delta$,*

$sign_{\geq}((SS^\top - nI_m)y) = S_{*j}$. *Here it is assumed that the coordinates $i$ at which $y_i \neq S_{ij}$ are chosen before $S$, or without knowledge of it.*

The $y^\top S_{*j}/\|y\| = y^\top S_{*j}/\sqrt{\|y\|_1}$ term may seem mysterious. $m - \|y\|_1$ is the number of erasures in $y$, that is, the number of zero coordinates, and $m - |y^\top S_{*j}|$ is the number of erasures plus twice the number of error coordinates $i$ where $y_i \neq S_{ij}$. When there are only erasures, and $y = S_{*j}$ except for errors, $\|y\|_1 = y^\top S_{*j}$, and $\|y\|_1 \geq 2n\log(2m/\delta)$ suffices.

We also introduce a variant of Hopfield nets, which we call Hopfield$\pm$, in which the vector outer products yielding $W$ are each multiplied by a random $\pm 1$ value before summing.[2] As described in Theorem 14 below, which establishes a JL-like property for Hopfield$\pm$, this system can store and recover $m^2$ (up to log factors) vectors. Considering that it uses $O(m^2)$ space, its storage efficiency for bundling is not far from that of MAP-I.

**Theorem 14.** *Given $\varepsilon, \delta \in (0,1]$, scaled sign matrix $\bar{S} \in \frac{1}{\sqrt{m}}\{\pm 1\}^{m \times d}$ with uniform independent entries, diagonal matrix $V \in \mathbb{R}^{d \times d}$, and diagonal matrix $D \in \{0, \pm 1\}^{d \times d}$ with uniform independent $\pm 1$ diagonal entries. There is $m = O(\varepsilon^{-1}\log(d/\delta)^2)$ such that with failure probability $\delta$, $\|\bar{S}VD\bar{S}^\top\|_F^2 = (1 \pm \varepsilon)\|V\|_F^2$.*

**Roadmap for Proofs.** Theorem 13 is proven in §C.1. Theorem 14 is discussed in §C.2.

### 3.3 ANALYSIS OF MAP-B

For MAP-B, we write our VSA atomic vectors as $Se_i$, where $S$ is a random sign matrix (Def. 1). However, we require here that even composite (non-atomic) VSA vectors be sign vectors, so a bundle of atomic vectors is represented as $\text{sign}(Sv)$ for a characteristic vector $v$.

The nonlinearity of bundling makes analysis more difficult. While we were able to reason about set intersection readily via the JL framework when analyzing MAP-I, our results for MAP-B only hold for testing set membership, which is a specific instance of set intersection. We are able to show that membership testing, to determine if $i \in [d]$ is in $\text{supp}(v)$, as represented by the bundle $x = \text{sign}(Sv)$, can be done by checking if $x^\top S_{*i} \geq \tau$, with threshold $\tau$ specified below.

**Theorem 15.** *For $v \in \{0,1\}^d$, let $x = \text{sign}(Sv)$ be the MAP-B bundling of $n = \|v\|_1$ atomic vectors. Then for all $i \in [m]$ and $j \in \text{supp}(v)$, $\Pr[x_i S_{ij} = +1] = 1/2 + \Theta(1/\sqrt{n})$, as $n \to \infty$, and there is $m = O(n\log(d/\delta))$ such that with failure probability $\delta$, $j \in [d]$ has $j \in \text{supp}(v)$ if and only if $x^\top S_{*j} = e_j^\top S\,\text{sign}(Sv)$ has $x^\top S_{*j} \geq \sqrt{2m\log(2d/\delta)}$.*

The proof borrows ideas from Boolean Fourier analysis, namely the analysis of the majority function, see O'Donnell (2014) for a thorough guide. In §D.1.3, we also provide an extension of Thm. 15 to testing for whether the set intersection between two sets $X, Y$ is empty or not.

We also remark that the bundling operator for MAP-B is *not* associative; while $\text{sign}(Sv)$ is one way that MAP-B might compute a bundle, in a more general setting, there could be a sequence of bundling operations. This would result in a tree of bundle operations of some depth greater than one, in contrast to $\text{sign}(Sv)$, whose tree has depth one. We prove that the reliability of the bundling operation decays exponentially with the depth.

**Lemma 16.** *Given independent sign vectors $x^{(i)} \in \{\pm 1\}^m$, $i \in [r]$, construct vector $x$ by setting $x \leftarrow x^{(1)}$, and for $j = 2, \ldots, r$, setting $x \leftarrow \text{sign}(x + x^{(j)})$. Then, for $\ell \in [m]$,*

$$\Pr[x_\ell^{(1)} x_\ell = 1] = 1/2 + 1/2^r$$

**Rotations and Bundles of Bindings:** In the Appendix §D, we analyze for MAP-B, rotations in the same setting described in Definition 7 and Theorem 8, and bundles of bindings in the setting of Definition 10 and Theorem 11.

---

[2]The name Hopfield$\pm$ references the Rademacher values, and is in homage to many variants of X called X++.

## 3.4 Sparse Binary Bundling and Bloom Filter Analysis

If we use VSA vectors $Bx$ with $B$ chosen as a sparse binary matrix (defined below), we obtain two new families of VSAs, which we refer to as *Bloom filters* and *Counting Bloom filters*, from their names in the data structures literature.

**Definition 17.** *For given $m$ and $k \leq m$, a $k$-sparse binary vector $y \in \{0, 1\}^m$ has $\|y\|_0 \leq k$, chosen by adding $k$ random natural basis vectors $e_i \in \{0, 1\}^m$, with each $i$ chosen independently and uniformly from $[m]$. A $k$-sparse binary matrix $B \in \{0, 1\}^{m \times d}$ has columns that are independent sparse binary vectors, and a scaled $k$-sparse binary matrix has the form $\bar{B} = \frac{1}{k}B$, where $B$ is a sparse binary matrix. These may be called just* sparse, *leaving the $k$ implicit.*

The Bloom and Counting Bloom filters slightly generalize BSDC-CDT, described in Schlegel et al. (2022). For more on Bloom filters, see Bloom (1970); Broder & Mitzenmacher (2004). Bloom filter represents the set $\mathrm{supp}(v)$ for $v \in \{0, 1\}^d$ as $1 \wedge Bv$, denoting the vector $x$ with $x_i = \min\{1, (Bv)_i\}$. We give VSA dimension bounds enabling reliable estimation of set intersections; a rigorous version of such an estimator appears to be novel for Bloom filters. We define a function $h_{m,k}(\cdot)$ that maps the expected dot product of the VSA bundles to the dot product of the original vectors.

**Theorem 18.** *Let $v, w \in \{0, 1\}^d$ represent sets $X, Y$ respectively, and define the following counts:*

$$n \equiv |X \cap Y| = v \cdot w, \qquad n_v \equiv |X \setminus Y| = \|v\|_0 - n, \qquad n_w \equiv |Y \setminus X| = \|w\|_0 - n$$

*Let $x = 1 \wedge Bv$ and $y = 1 \wedge Bw$, where $B \in \{0, 1\}^{m \times d}$ is a sparse binary matrix.*

*Assume WLOG $n_w \geq n_v$. Then for $\delta \in (0, 1)$ and $\varepsilon > 0$, there are $k = O(\varepsilon^{-1} \log(1/\delta))$ and $m = O(k\varepsilon^{-1}(n_v n_w + n^2 + \varepsilon(n + n_w)))$ such that $h_{m,k}(x \cdot y) = n \pm \epsilon$ with failure probability $\delta$. Here, $h_{m,k}(z)$ (see Lemma 41) maps the Bloom filter output $1 \wedge Bz$ to an estimate of $\|z\|_0$.*

We remark that $h_{m,k}(x \cdot y)$ is not an unbiased estimator of $v \cdot w$; we use a "Bernstein version" of McDiarmid (Theorem 24) to establish concentration of $x \cdot y$ first, and conclude the success of the estimator $h_{m,k}(x \cdot y)$ from there. In our analysis, we also divide $x \cdot y$ into two parts: the 1s contributed by $\mathrm{supp}(v) \cap \mathrm{supp}(w)$ after applying $B$ and the min operation, and the 1s contributed by $\mathrm{supp}(v)\Delta\,\mathrm{supp}(w)$, which may increment the same index after applying $B$.

In the *Counting Bloom filter*, the bundle is simply $\bar{B}v$. Here, we want to compute the *generalized set intersection* of two vectors $v, w$, which refers to the coordinate-wise minimum $v \wedge w$. The generalized set intersection size is $v \wedge w = \|v \wedge w\|_1$. Even though the bundle is a linear map, as was the case with MAP-I, we ultimately analyze a non-linear function for set similarity.

**Theorem 19.** *Let $v, w \in \mathbb{R}^d_{\geq 0}$. Let $K_b \equiv \|v - w\|_\infty \leq \max\{\|v\|_\infty, \|w\|_\infty\}$, and define:*

$$n \equiv v \wedge w, \qquad n_v \equiv \|v\|_1 - n, \qquad n_w \equiv \|w\|_1 - n,$$

*Let $x = Bv$ and $y = Bw$, where $B \in \{0, 1\}^{m \times d}$ is a sparse binary matrix. Then, for $\delta \in (0, 1)$ and $\varepsilon > 0$, there are $k = O(K_b \varepsilon^{-1} \log(1/\delta))$ and $m = 12\pi^2 k\varepsilon^{-1} n_v n_w = O(K_b \varepsilon^{-2} n_v n_w \log(1/\delta))$ such that*

$$(x \wedge y)/k - (v \wedge w) \in [0, \epsilon)$$

*with failure probability at most $\delta$. Since $n_v + n_w = \|v - w\|_1$, $m = O(K_b \varepsilon^{-2} \|v - w\|_1^2 \log(1/\delta))$ also suffices (using the AM-GM inequality on $n_v n_w$). If $\|v\|_1, \|w\|_1$ are stored, then $\|v - w\|_1$ can be estimated up to additive $\varepsilon$ with the same $m$.*

Our theorem helps analyze weighted sets, represented as $v, w \in \mathbb{Z}^d_{\geq 0}$, where $\mathbb{Z}_{\geq 0}$ denotes the non-negative integers. This also translates to a bound for estimating the $\ell_1$ distance between $v$ and $w$, via $\|v - w\|_1 = \|v\|_1 + \|w\|_1 - 2(v \wedge w)$. Similarly to our proof of Theorem 18, we split $\frac{1}{k}(x \wedge y)$ into a contribution related to $v \wedge w$, which is $\mathrm{supp}(v) \cap \mathrm{supp}(w)|$ for $v, w \in \{0, 1\}^d$, and a contribution from $v - (v \wedge w)$ and $w - (v \wedge w)$, which are the characteristic vectors of $\mathrm{supp}(v) \setminus \mathrm{supp}(w)$ and $\mathrm{supp}(w) \setminus \mathrm{supp}(v)$ for $v, w \in \{0, 1\}^d$. The proof of Theorem 19 also relies on the "Bernstein version" of McDiarmid's inequality (Theorem 24).

**Roadmap for Proofs:** In §E.1, we prove Theorem 18, and in §E.2, we prove Theorem 19.

**Concluding remarks.** In this paper, we have presented analyses of the capacity of four different VSA representations (and the novel Hopfield$\pm$ network) for various symbolic operations, and their relation to known concepts. Yet, much remains, including the analysis of VSAs where the atomic vectors have semantic content, as in distance sensitive Bloom filters (Kirsch & Mitzenmacher, 2006).

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

## A    PRELIMINARIES AND CONCENTRATION INEQUALITIES

In this section, we include a few useful concentration inequalities for functions of random variables that are used throughout our VSA analysis.

Let $f : \Omega_1 \times \ldots \times \Omega_n \to \mathbb{R}$ be an arbitrary function on $n$ variables.

**Definition 20.** *A function* $f : \Omega_1 \times \ldots \times \Omega_n \to \mathbb{R}$ *has* bounded differences *with constants* $\{c_i\}_{i \in [n]}$ *if for all* $x_1 \in \Omega_1, \ldots, x_n \in \Omega_n$ *and all* $i \in [n]$,

$$\sup_{y \in \Omega_i} |f(x_1, \ldots, x_{i-1}, x_i, x_{i+1}, \ldots, x_n) - f(x_1, \ldots, x_{i-1}, y, x_{i+1}, \ldots, x_n)| \leq c_i$$

When we want to study how $f$ applied to a random variable $X$ concentrates around its mean $\mathbb{E}[f(X)]$, the most standard tool is McDiarmid's inequality. The usual form of McDiarmid's inequality is the following statement:

**Theorem 21.** *Let* $X = (X_1, \ldots, X_n)$ *be a tuple of independent random variables supported on* $\Omega_1, \ldots, \Omega_n$, *respectively. If* $f$ *has bounded differences with constants* $c_1, \ldots, c_n$, *then:*

$$\Pr[f(X) - \mathbb{E}[f(X)] > t] \leq \exp\left(-\frac{2t^2}{\sum_{i=1}^n c_i^2}\right)$$

$$\Pr[f(X) - \mathbb{E}[f(X)] < -t] \leq \exp\left(-\frac{2t^2}{\sum_{i=1}^n c_i^2}\right)$$

We now derive a standard variant (Corollary 23) of McDiarmid that applies when the bounded differences hold with high probability. This is not a novel modification, but we include it for completeness. The corollary follows from the following lemma, which we will also use throughout our VSA analysis.

**Lemma 22.** *Let* $X$ *be a random variable taking values in domain* $D$, *e.g.* $D = \mathbb{R}^n$. *Let* $f$ *and* $g$ *map from* $D$ *to* $\mathbb{R}$, *such that* $\Pr[g(X) \neq f(X)] \leq \delta$, *with* $g(X) = 0$ *when* $g(X) \neq f(X)$, *and* $\sup_{x \in D} f(x) \leq M$ *for a value* $M \in \mathbb{R}$. *Then*

$$\Pr[f(X) - \mathbb{E}[f(X)] > t + \delta M] \leq \Pr[g(X) - \mathbb{E}[g(X)] > t] + \delta$$
$$\Pr[f(X) - \mathbb{E}[f(X)] < -t - \delta M] \leq \Pr[g(X) - \mathbb{E}[g(X)] < -t] + \delta$$

*Proof.* Let $B$ be the "bad subset" of $D$ where $g(X) \neq f(X)$. Here $\mathbb{E}[g(X)]$ is close but not equal to $\mathbb{E}[f(X)]$:

$$\mathbb{E}[f(X)] = \sum_{x \notin B} \Pr[X = x] \cdot f(x) + \sum_{x \in B} \Pr[X = x] \cdot f(x)$$

$$\leq \sum_{(x) \notin B} \Pr[X = x] \cdot g(x) + \delta M = \mathbb{E}[g(X)] + \delta M$$

Similarly, $\mathbb{E}[f(X)] \geq \mathbb{E}[g(X)] - \delta M$

Using this relationship between the means of $f(X)$ and $g(X)$, we have.

$$\Pr[f(X) - \mathbb{E}[f(X)] \geq t'] \leq \Pr[f(X) - \mathbb{E}[f(X)] \geq t' \mid f(X) = g(X)] \cdot \Pr[f(X) = g(X)] + \delta$$
$$\leq \Pr[g(X) - \mathbb{E}[f(X)] \geq t' \mid f(X) = g(X)] \cdot \Pr[f(X) = g(X)] + \delta$$
$$\leq \Pr[g(X) - \mathbb{E}[f(X)] \geq t'] + \delta$$
$$\leq \Pr[g(X) - \mathbb{E}[g(X)] \geq t' - \delta M] + \delta$$

The final line comes from the fact that $\mathbb{E}[f(X)] \geq \mathbb{E}[g(X)] - \delta M$. Letting $t' = t + \delta M$ yields the desired bound. The upper bound for $\Pr[f(X) - \mathbb{E}[f(X)] \leq -t']$ is analogous. ☐

**Corollary 23.** *Let* $X_1, \ldots, X_n$ *be independent random variables supported on* $\Omega_1, \ldots, \Omega_n$, *respectively. Define* $M := \sup_{x_1, \ldots, x_n} |f(x_1, \ldots, x_n)|$. *If with probability* $1 - \delta$ *over* $X_1, \ldots, X_n$, *the function* $f$ *has bounded differences with constants* $c_1, \ldots, c_n$, *i.e. there is a set* $S \subseteq \Omega$ *with* $\Pr[X \in S] = 1 - \delta$ *such that for all* $(x_1, \ldots, x_n) \in S$ *and* $i \in [n]$, *we satisfy*

$$\sup_{y \in \Omega_i} |f(x_1, \ldots, x_{i-1}, x_i, x_{i+1}, \ldots, x_n) - f(x_1, \ldots, x_{i-1}, y, x_{i+1}, \ldots, x_n)| \leq c_i$$

| VSA | Atomic Vector | Bundling |
|---|---|---|
| MAP-I | $\bar{S}e_i$ | $\bar{S}v$ |
| MAP-B | $Se_i$ | $\text{sign}(Sv)$ |
| Bloom filter | $Be_i$ | $1 \wedge Bv$ |
| Counting Bloom | $Be_i$ | $Bv$ |

Table 3: Equivalent to Table 1. Here $e_i \in \{0,1\}^d$ for $i \in [d]$ has $e_i = 1$, all other coordinates zero, and $v \in \{0,1\}^d$ represents the set $\text{supp}(v) \subset [d]$. The matrices $S$, $\bar{S}$, and $B$ are sign (also called bipolar or Rademacher), scaled sign, and sparse binary, as in Def. 1 and 17.

*then both of the following inequalities hold.*

$$\Pr\left[f(X) - \mathbb{E}[f(X)] > t + \delta M\right] \leq \exp\left(-\frac{2t^2}{\sum_{i=1}^n c_i^2}\right) + \delta$$

$$\Pr\left[f(X) - \mathbb{E}[f(X)] < -t - \delta M]\right] \leq \exp\left(-\frac{2t^2}{\sum_{i=1}^n c_i^2}\right) + \delta$$

*Proof.* Apply Lemma 22 to $f$ and the function $g$ defined as taking the same value of $f$ for $X_1, \ldots, X_n$ such that the bounded differences hold, and zero otherwise. Then, we apply Theorem 21 to $g$. $\qquad\square$

There is also a version of McDiarmid's inequality that incorporates variances; we refer to it as the "Bernstein form of McDiarmid's Inequality:"

**Theorem 24** (Ying (2004)). *Let $X = (X_1, \ldots, X_n)$ be a tuple of independent random variables supported on $\Omega \equiv \prod_{i \in [n]} \Omega_i$. For function $f : \Omega \to \mathbb{R}$ and $x \in \Omega, i \in [n], y \in \Omega_i$, let $g_i(x, y) \equiv f(x_1, \ldots, x_{i-1}, y, x_{i+1}, \ldots, x_n)$. (Note that $g_i(x, y)$ does not depend on $x_i$.) Let*

$$\textit{Var}_i(f) \equiv \sup_{x \in \Omega} \mathbb{E}_{X_i}\left[\left(g_i(x, X_i) - \mathbb{E}_{X_i'} g_i(x, X_i')\right)^2\right]$$

$$\tilde{\sigma}(f) \equiv \sum_{i \in [n]} \textit{Var}_i(f)$$

$$B(f) \equiv \max_{i \in [n]} \sup_{x \in \Omega} \left|g_i(x, X_i) - \mathbb{E}_{X_i'} g_i(x, X_i')\right|$$

*Then*

$$\Pr\left[f(X) - \mathbb{E}[f(X)] > t\right] \leq \exp\left(-\frac{2t^2}{\tilde{\sigma}(f) + tB(f)/3}\right)$$

$$\Pr\left[f(X) - \mathbb{E}[f(X)] < -t\right] \leq \exp\left(-\frac{2t^2}{\tilde{\sigma}(f) + tB(f)/3}\right)$$

We will also need this known concentration bound for Rademacher random variables.

**Theorem 25.** *Let $X_1, \ldots, X_n$ be independent Rademacher random variables. Let $(a_1, \ldots, a_n)$ be an arbitrary vector in $\mathbb{R}^n$. Then,*

$$\Pr\left[\sum_{i=1}^n a_i X_i \geq t\|a\|_2\right] \leq \exp\left(-\frac{t^2}{2}\right)$$

A correspondence between the VSAs we study and their choice of matrix $P$ is found in Table 3.

## B   ANALYSIS OF MAP-I USING JOHNSON-LINDENSTRAUSS

In §B.1, we analyze MAP-I bundling, including the relation of norm estimation to distance, set intersection, and angle estimation, and discuss how the JL perspective extends to the sparse JL and Subsampled Randomized Hadamard Transforms (SRHTs).

Next, in §B.2, we analyze the VSA dimension needed to reliably estimate the intersection of two *sequences* of sets, as encoded using rotations, in both a general and a more restricted setting. This entails proving the JL property for a specific kind of random matrix with dependent entries.

Finally, in §B.3, we analyze intersection of bundles of bound pairs and bundles of $k$-wise bindings. Again, we establish the JL property for a random matrix with dependent entries. While bundles of key-value bindings have been analyzed before, bundles of $k$-wise bindings have not.

### B.1 BUNDLING AND SET INTERSECTION

The results in this section may apply to VSAs beyond just MAP-I where the bundling operator is addition over the real numbers. Section 3.1 of Thomas et al. (2021a) gives some representational bounds in that setting. Though we provide similar bounds, we frame our results using the Johnson-Lindenstrauss (JL) random projection lemma, which provides a simpler yet powerful lens through which to view the MAP-I VSA. (A connection to JL was briefly mentioned by Kent (2020), but it did not factor into any capacity analysis there.)

We first introduce the Johnson-Lindenstrauss (JL) lemma:

**Lemma** (Restatement of Lemma 2). *Suppose $\bar{S} \in \frac{1}{\sqrt{m}}\{-1, 1\}^{m \times d}$ is a scaled sign matrix (described in Def. 1). Then for given $\delta, \epsilon > 0$ there is $m = O(\epsilon^{-2}\log(1/\delta))$ such that for given vector $v \in \mathbb{R}^d$, it holds that $\|\bar{S}v\| = \|v\|(1 \pm \epsilon)$, with failure probability at most $\delta$.*

**Remark 26.** *We have a similar result for any matrix $P$ with i.i.d. subGaussian entries.*

This approximation bound can be easily translated to a bound for multiple differences of vectors. The proof is simply the application of of Lemma 2 to the $O(n^2)$ pairs of differences of vectors of $\mathcal{V}$, and a union bound (yielding the $\log n$ part of the bound).

**Corollary** (Restatement of Corollary 3). *If random matrix $P$ satisfies the JL property (Lemma 2), then for given $\delta, \epsilon > 0$, and a set $\mathcal{V} \subset \mathbb{R}^d$ of cardinality $n$, there is $m = O(\epsilon^{-2}\log(n/\delta))$ such that with failure probability $\delta$, for all pairs $v, w \in \mathcal{V}$, $\|P(v - w)\|^2 \in \|v - w\|^2(1 \pm \epsilon)$.*

In other words, the linear mapping $P$ preserves Euclidean distances up to a multiplicative factor of $(1 \pm \epsilon)$. If $v$ and $w$ are characteristic vectors for sets $X$ and $Y$, then $\|v - y\|^2 = |X\Delta Y|$, the cardinality of the symmetric difference of $X$ and $Y$. Thus, by taking the difference between their VSA representations $Pv$ and $Pv$, we can estimate how much the sets $X$ and $Y$ disagree, with small *relative* error $\epsilon$ in that estimate, mild dependence on the number of sets we can represent in this way, and no dependence, in the error, on the groundset size $n$.

However, this doesn't say what the needed dimensionality $m$ is, for accurate estimation of the size of the intersection. We consider this next, using the following standard corollary of Lemma 2.

**Corollary** (Restatement of Corollary 4). *Suppose the random matrix $P$ satisfies the JL property (Lemma 2). Then for $v, w \in \mathbb{R}^d$, there is $m = O(\epsilon^{-2}\log(1/\delta))$ so that $v^\top P^\top P w = v^\top w \pm \epsilon\|v\|\|w\|$ with failure probability $\delta$.*

*Proof.* First, suppose $v$ and $w$ are unit vectors. By Lemma 2 and a union bound, with failure probability at most $3\delta$ we have $v^\top P^\top P v = v^\top v \pm \epsilon$, $w^\top P^\top P w = w^\top w \pm \epsilon$, and $(v+w)^\top P^\top P(v+w) = (v + w)^\top(v + w) \pm 2\epsilon$. Since $2v^\top w = (v + w)^\top(v + w) - v^\top v - w^\top w$ and matrix multiplication is an application of a linear operator, an analogous expression holds for $v^\top P^\top P w$. Then,

$$2(v^\top P^\top P w - v^\top w) = \pm\epsilon \pm \epsilon \pm 2\epsilon = \pm 4\epsilon,$$

with failure probability $3\delta$. If $v$ and $w$ are not unit vectors, we apply this reasoning to $v/\|v\|$ and $w/\|w\|$, and then multiply through by $\|v\|\|w\|$, getting $v^\top P^\top P w = v^\top w \pm 2\epsilon\|v\|\|w\|$. That is, the conclusion holds with relative error parameter $2\epsilon$ and failure probability $3\delta$. We can fold the factors 2 and 3 into the bound for $m$, and the result follows. $\square$

Beyond the approximation condition, the proof only depends on $P$ being a linear operator. Suppose $P$ is a scaled sign matrix $\bar{S}$, and let $v, w \in \{0, 1\}^d$ be characteristic vectors. We want $v^\top P^\top P w = v^\top w \pm \epsilon_0$, where $\epsilon_0 < 1/2$. Since $v^\top w$ is an integer, this assures that $v^\top P^\top P w$ rounds to $v^\top w$. The next lemma gives conditions on $\epsilon$ and $\delta$ to attain this.

**Theorem** (Restatement of Theorem 5). *Suppose random matrix $P$ satisfies the JL property (Lemma 2). Given $M$ pairs of characteristic vectors $v, w \in \{0,1\}^d$ such that for every pair $v, w$, $\|v\|_1 \|w\|_1 \leq N$, then there is $m = O(N \log(M/\delta))$ such that $\lfloor v^\top P^\top P w \rceil = v^\top w$ for all $M$ pairs with probability $\geq 1 - \delta$.*

**Remark 27.** *In particular, this holds for the following special cases.*

- ***Set membership**: $v$ is a characteristic vector, and $w$ is a standard basis vector $e_i$ for $i \in [d]$. Then, $N = \|v\|_1$, $M = d$, and VSA dimension $m = O(\|v\|_1 \log(d/\delta))$ suffices.*

- *To get all pairwise intersections among $M^{O(1)}$ vectors $v$ with $\|v\|_1 \leq \sqrt{N}$, we can use $m = O(N \log(M/\delta))$.*

*Proof.* For a given pair of vectors $v$ and $w$, by Corollary 4 it is enough to choose $\epsilon_0 < 1/2$ and $\epsilon = \epsilon_0/(\|v\|\|w\|) \geq \epsilon_0/\sqrt{N}$, to achieve the desired accuracy.

If $\delta'$ is the failure probability of Corollary 4, then the overall failure probability is at most $M\delta'$, by a union bound. Thus for $\delta' = \delta/M$ and $\epsilon$ from the JL property equal to $\epsilon_0/\sqrt{N}$, the dimension $m$ of Corollary 4 is $m = O(N \log(M/\delta))$, and every dot product estimate is accurate to within additive $\epsilon_0$ with failure probability at most $\delta$. The result follows. $\square$

**Angle estimation**     Let $\Theta_{v,w}$ denote the angle between vectors $v$ and $w$.

**Lemma 28.** *Under the conditions of Lemma 2, for $v, w \in \mathbb{R}^d$, there is $m = O(\epsilon^{-2} \log(1/\delta))$ so that $\cos \Theta_{\bar{S}v, \bar{S}w} = \cos \Theta_{v,w} \pm \epsilon$ with failure probability $\delta$.*

*Proof.* This follows from $\cos \Theta_{v,w} = \frac{v^\top w}{\|v\|\|w\|}$, and similarly for $\Theta_{\bar{S}v, \bar{S}w}$; applying Corollary 4 to the estimation of $v^\top w$, $\|v\|$, and $\|w\|$, and finally folding constant-factor increases in $\epsilon$ and $\delta$ into $m$. $\square$

### B.1.1 Variations Using Alternative Sketching Matrices

**Using sparse JL**     The sparse JL transform has $\epsilon m$ nonzero entries per column of $P$, each entry an independent sign (that is, $\pm 1$). Such a matrix, multiplied by a scaling factor, also satisfies Lemma 2, as shown in Kane & Nelson (2014); Cohen et al. (2018), and therefore could be used in VSAs with the same performance, at least as far as bundling goes. However, element-wise product does not seem to be enough to do binding effectively for such a representation, as the element-wise product of sparse vectors is likely to be all zeros.

**Using Subsampled Randomized Hadamard Transform (SRHT)**     The SRHT, described in Boutsidis & Gittens (2013), is multiplication by a matrix $ZHD$, where $D$ is a random sign diagonal matrix (random sign flips on the diagonal), $H$ is a Hadamard matrix (an orthogonal sign matrix), and $Z$ simply selects a uniform random subset of the rows (a diagonal binary matrix). This operation, along with a scaling factor, is shown by Krahmer & Ward (2011) to satisfy the conditions of Lemma 2, up to needing the larger target dimension $m = O(\epsilon^{-2} \log(1/\delta) \log^4 d)$, that is, with an additional factor of $\log^4 d$. While for many applications, a useful property of SRHT is that $ZHDv$ is rapidly computed, here the main attraction is that the generation or storage of random sign values is greatly reduced; entries of the $H$ matrix are readily computed using bitwise operations on the entry indices.

### B.2 Rotations for Encoding Sequences

For scaled sign matrices, using a rotation (i.e. a permutation that is a single large cycle), we can store length-$L$ sequences of vectors with an error bound that grows quadratically with $L$. It is possible that there is a better general bound than this; we found better performance in a special case. If the vectors are, in particular, characteristic vectors, so that a sequence of symbols is being stored (and not e.g. a sequence of embeddings), then better error behavior (that depends on how much the vectors overlap, rather than naively on $L$) typically occurs.

Before further definition and analysis, we have some lemmas useful for the analysis.

**Lemma 29.** *Say we sample independent scaled random sign matrices $P_1, P_2 \in \{\pm 1\}^{m \times d}$ with $m$ large enough to satisfy the JL property, i.e. for given $v \in \mathbb{R}^d$, with failure probability $\delta$, it holds that $\|P_i x\|^2 = (1 \pm \epsilon)\|x\|^2$. If the block matrix $[P_1 \, P_2]$ also satisfies the JL property, then for given unit vectors $x_1, x_2 \in \mathbb{R}^d$, with failure probability at most $3\delta$, $|x_1^\top P_1^\top P_2 x_2| \leq 2\epsilon$.*

*Proof.* Since $P_1$ and $P_2$ were sampled independently, the block matrix $[P_1 \, P_2]$ itself is a larger random sign matrix, and it satisfies the JL property for $\varepsilon$ with probability $\leq \delta$. We have

$$\|P_1 x_1 + P_2 x_2\|^2 = \|[P_1 \, P_2] \begin{bmatrix} x_1 \\ x_2 \end{bmatrix}\|^2 = (\|x_1\|^2 + \|x_2\|^2)(1 \pm \epsilon) = 2(1 \pm \epsilon)$$

with failure probability $\delta$, since $[P_1 \, P_2]$ satisfies the same conditions as $P_1, P_2$. With failure probability at most $3\delta$, the embedding conditions hold for $[P_1 \, P_2]$, $P_1$, and $P_2$. Assuming they do,

$$|x_1^\top P_1^\top P_2 x_2| = \left| \frac{1}{2} \left[ \|P_1 x_1 + P_2 x_2\|^2 - \|P_1 x_1\|^2 - \|P_2 x_2\|^2 \right] \right|$$

$$\leq \frac{1}{2} \left[ 2(1 + \epsilon) - (1 - \epsilon) - (1 - \epsilon) \right] \leq 2\epsilon.$$

The result follows. $\qquad \square$

**Lemma 30.** *Let $R \in \mathbb{R}^{m \times m}$ denote a rotation matrix, as in Def. 7. Let $\bar{S} \in \mathbb{R}^{m \times d}$ be a scaled sign matrix such that for given $x \in \mathbb{R}^d$, $\|\bar{S}x\|^2 = \|x\|^2 (1 \pm \epsilon)$ with failure probability $\delta$. Then for unit vectors $x, y \in \mathbb{R}^d$ and $r \leq m/2$, we have $|x^\top \bar{S}^\top R^r \bar{S} y| \leq 4\epsilon$, with failure probability $6\delta$. If $x$ and $y$ are not unit vectors, then $|x^\top \bar{S}^\top R^r \bar{S} y| \leq 4\epsilon \|x\| \|y\|$.*

*Proof.* We have $(R^r \bar{S})_{i*} = \bar{S}_{[(i+r-1)\%m+1]*}$, where $j\%m$ denotes $j$ modulo $m$. Consider the division of the rows of $\bar{S}$ and of $R^r \bar{S}$ into blocks of size $r$. The first $r$ rows of $\bar{S}$ and the first $r$ rows of $R^r \bar{S}$ comprise together the first and second blocks of $\bar{S}$, since $\{(i + r - 1)\%m + 1 | i \in [r]\} = \{1 + r, 2 + r, \ldots, r + r\}$. The second block of $r$ rows of $\bar{S}$ and of $R^r \bar{S}$ comprise together the second and third block of rows of $\bar{S}$, and so on.

As a consequence, $\bar{S}$ and $R^r \bar{S}$ can each be split into two matrices $\bar{S}_1, \bar{S}_2$ and $(R^r \bar{S})_1, (R^r \bar{S})_2$, where $\bar{S}_1$ comprises the odd-numbered blocks of $\bar{S}$, and $(R^r \bar{S})_1$ comprises the corresponding blocks of $R^r \bar{S}$, such that the rows of $\bar{S}_1$ and the rows of $(R^r \bar{S})_1$ are chosen from disjoint sets of rows of $\bar{S}$, and similarly for $\bar{S}_2$ comprising the even-numbered blocks of $\bar{S}$, and $(R^r \bar{S})_2$ comprising the corresponding blocks of $R^r \bar{S}$. That is, splitting the entries of $x$ and $y$ in the same way into vectors $x_1, x_2, y_1, y_2$, we have that

$$x^\top \bar{S}^\top R^r \bar{S} y = x_1^\top \bar{S}_1^\top (R^r \bar{S})_1 y_1 + x_2^\top \bar{S}_2^\top (R^r \bar{S})_2 y_2.$$

Since the entries of $\bar{S}_1$ were chosen independently from those of $(R^r \bar{S})_1$, we can apply Lemma 29 to have that $|x_1^\top \bar{S}_1^\top (R^r \bar{S})_1 y_1| \leq 2\epsilon$, with failure probability $C\delta$, with a similar statement for $x_2^\top \bar{S}_2^\top (R^r \bar{S})_2 y_2$. Adding these bounds, and taking a union bound on the failure probability, yields the result for unit vectors. The claim about non-unit vectors follows immediately. $\qquad \square$

**Definition 31.** *For $v \in \mathbb{R}^{Ld}$ encoding a sequence $v_{(0)}, v_{(1)}, \ldots v_{(L-1)} \in \mathbb{R}^d$, as in Def. 7, define*

$$\|v\|_{L,1} \equiv \sum_{0 \leq j < L} \|v_{(j)}\|$$

**Theorem** (Restatement of Theorem 8). *Given scaled sign matrix $\bar{S} \in \mathbb{R}^{m \times d}$, rotation matrix $R \in \mathbb{R}^{m \times m}$ as in Def. 6, integer $L > 0$, and $\bar{S}_{R,L}$ as in Def. 7. Then for $v \in \mathbb{R}^{Ld}$ as defined above,*

$$\left| \|\bar{S}_{R,L} v\|^2 - \|v\|^2 \right| \leq 3\epsilon \|v\|_{L,1}^2 \leq 3L\epsilon \|v\|^2,$$

*with failure probability $6L^2 \delta$. It follows that there is $m = O((L/\varepsilon)^2 \log(L/\delta))$ such that with failure probability at most $\delta$, $\|\bar{S}_{R,L} v\|^2 = (1 \pm \varepsilon)\|v\|^2$.*

Note that bounds for set difference and intersection follow by slight variation of Corollaries 3 and 4, and sharper ones using $\|v\|_{L,1}$ instead of $\|v\|$.

*Proof.* Recall that $R^\top R = I$ and $R^\top = R^{-1}$. Assuming the approximation bounds of Lemma 30 hold for several instances, we have that

$$
\begin{aligned}
\|\bar{S}_{R,L} v\|^2 &= \|\sum_{0 \le j < L} R^j \bar{S} v_{(j)}\|^2 \\
&= \sum_{0 \le j, j' < L} v_{(j')}^\top \bar{S}^\top (R^\top)^{j'} R^j \bar{S} v_{(j)} \\
&= \sum_{0 \le j < L} v_{(j)}^\top \bar{S}^\top \bar{S} v_{(j)} + 2 \sum_{0 \le j' < j < L} v_{(j')}^\top \bar{S}^\top R^{j-j'} \bar{S} v_{(j)} \\
&\le \|v\|^2 (1 \pm \epsilon) + 2 \sum_{0 \le j' < j < L} \epsilon \|v_{(j')}\| * \|v_{(j)}\| \\
&\le \|v\|^2 + 3\epsilon \left( \sum_{0 \le j < L} \|v_{(j)}\| \right)^2,
\end{aligned}
$$

as claimed. Here we use that for vector $z \in \mathbb{R}^L$, $\|z\|_2 \le \|z\|_1 \le \|z\|_2 \sqrt{L}$. This also implies the last inequality of the theorem statement. The failure probabilities of Corollary 4 for the diagonal terms, and Lemma 30 for the off-diagonal terms, imply the overall bound via a union bound. The last line follows using Lemma 2. □

### B.2.1 TIGHTER BOUNDS FOR CHARACTERISTIC VECTORS

The above theorem applies to any sequence of vectors, but better bounds are obtainable for characteristic vectors, in some cases: consider $v$ and $w$ the characteristic vectors of distinct symbols. Then $\bar{S}v$ and $R\bar{S}w$ are entry-wise independent, because $v$ and $w$ select different columns of $\bar{S}$, and the rotation $R\bar{S}$ doesn't change this. Next we apply this idea more generally.

**Theorem** (Restatement of Theorem 9). *Given scaled sign matrix $\bar{S} \in \mathbb{R}^{m \times d}$, rotation matrix $R \in \mathbb{R}^{m \times m}$ (Def. 6), integer $L > 0$, and $\bar{S}_{R,L}$ as in Def. 7. For a sequence of vectors $v_{(0)}, v_{(1)}, \dots v_{(L-1)} \in \{0,1\}^d$, let $K \equiv \|\sum_{0 \le j < L} v_{(j)}\|_\infty$. There is $m = O\left(K^2 \varepsilon^{-2} \log(K/(\varepsilon \delta))\right)$ such that with failure probability $\delta$,*

$$
|\|\bar{S}_{R,L} v\|^2 - \|v\|^2| \le \epsilon \|v\|^2
$$

Note that here, each $v_{(j)}$ is the characteristic vector of a collection of symbols, and $K$ is the maximum number of times that some symbol is represented, adding up appearances over the given vectors. Bounds for set difference and intersection follow by slight variation of Corollaries 3 and 4.

*Proof.* Our proof of this lemma uses a corollary of McDiarmid's Inequality (Theorem 21). The product $\bar{S}_{R,L} v$ is a sum of vectors of the form $R^j \bar{S}_{*i}$, for entries $i$ of $v_{(j)}$ equal to one. Recall that $v_{(j)} \in \mathbb{R}^d$ is the $j$-th block of $v$, and that we defined $K = \|\sum_{0 \le j < L} v_{(j)}\|_\infty$. We can decompose $v$ into a sum $v = \sum_{k \in [K]} x^k$, where each $x^k \in \{0,1\}^{Ld}$, such that no $j, j' \in \text{supp}(x^k)$ has $j \mod d = j' \mod d$. For each $k \in K$, the vector $\sqrt{m} \bar{S}_{R,L} x^k$ is a sum of independent sign vectors, since each entry of 1 in $x^k$ will select a different column of $\bar{S}$, and so Lemma 2 applies, and $\|\bar{S}_{R,L} x^k\|^2 = (1 \pm \epsilon) \|x^k\|^2$ with failure probability $\delta$.

We will show next for $k, k' \in [K]$ with $k \ne k'$, that $D_{k,k'} \equiv x^{k' \top} \bar{S}_{R,L}^\top \bar{S}_{R,L} x^k$ concentrates around its expectation $x^{k' \top} x^k$. To do this, we will apply McDiarmid's Inequality, (Corollary 23). First, we note that each entry $(\bar{S}_{R,L} x^k)_i$ is a sum of independent Rademacher values, scaled by $1/\sqrt{m}$, and so by Theorem 25 its square is at most $\frac{1}{m} \|x^k\|^2 \log(2/\delta_1)$ with failure probability $\delta_1$, for $\delta_1$ to be determined.

Let $\mathcal{E}$ be the event that all entries of $\bar{S}_{R,L} x^k$ and $\bar{S}_{R,L} x^{k'}$ satisfy this bound; this holds with failure probability at most $2\delta_1 m$. To apply Theorem 21, we need to bound, for each entry of $\bar{S}$, the change in $D_{k,k'}$ that can occur when the entry takes one of its values $+1$ or $-1$. Due to the construction of $x^k$ and $x^{k'}$, the entry affects only at most one entry $(\bar{S}_{R,L} x^k)_i$ and at most one entry $(\bar{S}_{R,L} x^{k'})_{i'}$.

The difference in $D_{k,k'}$ due to a change in an entry of $\bar{S}$ contributing to $(\bar{S}_{R,L}x^k)_i$ is at most $\frac{2}{\sqrt{m}}(\bar{S}_{R,L}x^{k'})_i$; if event $\mathcal{E}$ holds, its square is at most $\frac{4}{m^2}\|x^{k'}\|^2 \log(2/\delta_1)$. Similarly, the difference due to a change in an entry contributing to $\bar{S}_{R,L}x_i^{k'}$ is at most $\frac{4}{m^2}\|x^k\|^2 \log(2/\delta_1)$. Putting these together, the squared change in $D_{k,k'}$ due to an entry that appears on both sides is at most

$$(2|(\bar{S}_{R,L}x^{k'})_i| + 2|(\bar{S}_{R,L}x^k)_{i'}|)^2$$
$$= 4\left(|(\bar{S}_{R,L}x^{k'})_i|^2 + 2|(\bar{S}_{R,L}x^{k'})_i \cdot (\bar{S}_{R,L}x^k)_{i'}| + |(\bar{S}_{R,L}x^k)_{i'}|^2\right)$$
$$\leq 8|(\bar{S}_{R,L}x^{k'})_i|^2 + 8|(\bar{S}_{R,L}x^k)_{i'}|^2$$
$$= \frac{8}{m^2} \cdot \log(2/\delta_1)(\|x^{k'}\|^2 + \|x^k\|^2).$$

The third line follows from the AM-GM inequality.

To obtain the total of the squared difference bounds, as needed in Corollary 23, note that a given entry of $\bar{S}$ may be one of $m\|x^k\|^2$ relevant entries of $\bar{S}$ on one side, and may be one of $m\|x^{k'}\|^2$ relevant entries on the other. Suppose we "charge" an appearance of an entry among the $m\|x^k\|^2$ entries contributing to $\bar{S}_{R,L}x^k$ an amount of $\frac{8}{m^2}\log(2/\delta_1)\|x^{k'}\|^2$, and an appearance of an entry among among the $m\|x^{k'}\|^2$ contributing to $\bar{S}_{R,L}x^{k'}$ an amount of $\frac{8}{m^2}\log(2/\delta_1)\|x^k\|^2$. Then from the above, the contribution of every entry, and its one or two appearances, to the sum of squares of per-entry changes to $D_{k,k'}$, is accounted for. That sum of squares, as needed for Cor. 23, is therefore at most

$$\frac{16}{m}\log(2/\delta_1)\|x^k\|^2\|x^{k'}\|^2,$$

assuming still event $\mathcal{E}$.

To apply Cor. 23, we have upper bound $M = \|x^k\|^2\|x^{k'}\|^2$, and $\delta$ of that corollary at most $2\delta_1 m$. We note that since $k \neq k'$, we have $\mathbb{E}[D_{k,k'}] = 0$. So

$$\Pr(D_{k,k'} \geq t + 2\delta_1 m\|x^k\|^2\|x^{k'}\|^2) \leq 2\exp\left(-\frac{t^2 m}{16\log(2/\delta_1)\|x^k\|^2\|x^{k'}\|^2}\right) + 2\delta_1 m.$$

Setting $\delta_1 \leq \varepsilon\delta/4K^2 m$ for target failure probability $\delta$, and $t = \frac{1}{2}\varepsilon\|x^k\|\|x^{k'}\|$, we get

$$\Pr(D_{k,k'} \geq \varepsilon\|x^k\|^2\|x^{k'}\|^2) \leq 2\exp\left(-\frac{\varepsilon^2 m}{16\log(2/\delta_1)}\right) + \varepsilon\delta/2K^2,$$

so the failure probability is less than $\delta/K^2$ for $m = O(\varepsilon^{-2}\log(K/\varepsilon\delta))$. (As is not hard to verify.)

Suppose we have this bound for all $k \neq k'$, which by union bound holds with failure probability at most $\delta$. We have

$$\|\bar{S}_{R,L}v\|^2 = \sum_k (1 \pm \varepsilon)\|x^k\|^2 \pm \frac{1}{2}\varepsilon \sum_{k \neq k'} \|x^k\|\|x^{k'}\|$$

$$= \|v\|^2(1 \pm \varepsilon) \pm \varepsilon\left(\sum_k \|x^k\|\right)^2 = \|v\|^2(1 \pm \varepsilon) \pm K\varepsilon\|v\|^2,$$

and the theorem follows, after folding $K$ into $\varepsilon$ and adjusting constants. $\qquad\square$

## B.3 BINDING

### B.3.1 BUNDLES OF PAIR-WISE BINDINGS

As discussed in the introduction, a bundle of pairwise bindings can be given as a vector $\bar{S}^{\odot 2}v$, where $v \in \{0,1\}^{\binom{d}{2}}$, and $\bar{S}^{\odot 2} \in \{\pm\frac{1}{\sqrt{m}}\}^{m \times \binom{d}{2}}$ is defined in Def. 10. We have $\mathbb{E}[\|\bar{S}^{\odot 2}v\|^2] = \|v\|^2$, and need to show that $\|\bar{S}^{\odot 2}v\|^2 \in \|v\|^2(1 \pm \varepsilon)$ with high probability.

**Theorem** (Restatement of Theorem 11). *For $v \in \{0,1\}^{\binom{d}{2}}$, there is $m = O\left(\varepsilon^{-2}\log^3(\|v\|_1/\varepsilon\delta)\right)$ so that for $\bar{S}^{\odot 2} \in \{\pm\frac{1}{\sqrt{m}}\}^{\binom{d}{2}}$, we have $\Pr[|\|\bar{S}^{\odot 2}v\|^2 - \|v\|^2| > \varepsilon\|v\|^2] \leq \delta$.*

For our analysis we will use a graph $G(V, E)$ of interactions in $\bar{S}^{\odot k}$ of columns in $\bar{S}$. Index the columns of $\bar{S}^{\odot k}$ by the columns of $\bar{S}$ from which they came, so $\bar{S}^{\odot k}_{*, \{i_1, i_2, \ldots, i_k\}}$ denotes the column of $\bar{S}^{\odot k}$ that is the binding of columns $\bar{S}_{*, i_1}, \bar{S}_{*, i_2}, \ldots \bar{S}_{*, i_k}$ of $\bar{S}$. Also use this indexing scheme for $v \in \mathbb{R}^{\binom{d}{k}}$. Now for given $v \in \mathbb{R}^{\binom{d}{k}}$, the graph (or hyper-graph, for $k > 2$) $G$ has edges $E = \{e \subset [\binom{d}{k}] \mid v_e \neq 0\}$, and vertices $V = \{i \mid i \in e \in E\}$.

We can also include singleton atomic vectors in the bundling, by including the all-ones vector $\vec{1}_m$ as an atomic vector, bound to each singleton. Note that approximate orthogonality holds also for these bindings, and the dimension is $\binom{d+1}{2}$, within a constant factor of $\binom{d}{2}$.

We prove Theorem 11 using this graphical view of $\bar{S}^{\odot 2}v$ in the following way. For given $\ell \in [m]$ we can write the Rademacher variables contributing to nonzero summands of $\bar{S}^{\odot 2}_\ell v$ as $X_\ell = (x_1, x_2, \ldots, x_{|V|})$, for $V$ the vertex set of the associated graph $G$. Also, we can express $\bar{S}^{\odot 2}_\ell v$ as a function

$$\sqrt{m} \cdot \bar{S}^{\odot 2}_{\ell, *} v = f(X_\ell), \text{ where } f(x) = \sum_{\{i,j\} \in E} x_i x_j. \tag{1}$$

*Proof of Theorem 11.* For $i \in [|V|]$, let function $g_i(x_1, \ldots, x_{|V|})$ take value $x_i(1_{\{i\} \in E} + \sum_{j: (i,j) \in E} x_j)$. (Here $1_{\{i\} \in E}$ equals one when $\{i\} \in E$, and zero otherwise.) Then, for any $\ell \in [m]$, flipping the sign of $x_i$ changes any $f(X_\ell)$ by no more than $2|g_i(X_\ell)|$. By Theorem 25, with $a_1, \ldots, a_{\deg_G(i)} = 1$ and $t = \sqrt{6 \log(|E|m/\delta)}$, we have

$$\Pr\left[2|g_i(X_\ell)| > 2\sqrt{6 \deg_G(i) \cdot \log(2|E|m/\delta)}\right] < \frac{\delta^3}{8|E|^3 m^3}. \tag{2}$$

Let $\mathcal{E}$ be the event that these upper bounds hold for all $\ell \in [m], i \in V$. Then taking the union bound over all $m|V|$ such $g_i(X_\ell)$, event $\mathcal{E}$ occurs with failure probability at most

$$\delta_{\mathcal{E}} \equiv \frac{\delta^3}{4|E|^2 m^2},$$

noting that $2|E| \geq |V|$, since each vertex occurs in some edge, and at most two vertices occur in one edge.

For $\ell \in [m]$, let function $\check{f}(X_\ell)$ be equal to $f(X_\ell)$ when $\mathcal{E}$ holds, and be zero otherwise. By McDiarmid's Inequality, Theorem 21,

$$\Pr[|\check{f}(X_\ell) - \mathbb{E}[\check{f}(X_\ell)]| \geq t] \leq 2 \exp\left(\frac{-2t^2}{\sum_{i \in [|V|]} (2\sqrt{6 \deg_G(i) \cdot \log(|E|m/\delta)})^2}\right)$$
$$= 2 \exp\left(-\frac{t^2}{12|E| \log(|E|m/\delta)}\right).$$

From Proposition 2.5.2 of Vershynin (2018), therefore, for all $\ell \in [m]$, $\check{f}(X_\ell) - \mathbb{E}[\check{f}(X_\ell)]$ is a sub-Gaussian random variable, with proxy variance within a constant factor of at most

$$\sigma_{\check{f}}^2 \equiv 12|E| \log(|E|m/\delta).$$

Let $\bar{f} \equiv \mathbb{E}[\check{f}(X)]$, for Rademacher $X$. Since $\mathbb{E}[f(X)] = 0$, we have

$$|\bar{f}| = |\mathbb{E}[\check{f}(X) \mid \mathcal{E}] \Pr[\mathcal{E}] + \mathbb{E}[\check{f}(X) \mid \neg\mathcal{E}] \Pr[\neg\mathcal{E}]|$$
$$= |\mathbb{E}[f(X) \mid \mathcal{E}] \Pr[\mathcal{E}] + 0|$$
$$= |\mathbb{E}[f(X)] - \mathbb{E}[f(X) \mid \neg\mathcal{E}] \Pr[\neg\mathcal{E}]|$$
$$\leq |0 - \mathbb{E}[f(X) \mid \neg\mathcal{E}]\delta_{\mathcal{E}}|$$
$$\leq |E|\delta_{\mathcal{E}}, \tag{3}$$

since $|f(X)| \leq |E|$.

Let $\check{F} \equiv \frac{1}{m}\sum_\ell \check{f}(X_\ell)^2$, and $\check{F}_c \equiv \frac{1}{m}\sum_\ell (\check{f}(X_\ell) - \bar{f})^2$. Using Exercise 2.7.10 and Corollary 2.8.3 of Vershynin (2018), it follows that

$$\Pr[|\check{F}_c - \mathbb{E}[\check{F}_c]| > t] \le 2\exp\left(-cm\min\left(\frac{t^2}{\sigma_{\check{f}}^4}, \frac{t}{\sigma_{\check{f}}^2}\right)\right).$$

For $t = \varepsilon|E|/2 < \sigma_{\check{f}}^2$, the upper bound is $2\exp(-cm\varepsilon^2/\log(|E|m/\delta)^2)$, for a different constant $c$, so there is $m = O(\varepsilon^{-2}\log(|E|/\varepsilon\delta)^3)$ such that

$$\Pr[|\check{F}_c - \mathbb{E}[\check{F}_c]| > \varepsilon|E|/2] \le \delta/2. \tag{4}$$

We have a tail estimate for $\check{F}_c$, but not for $\check{F}$ or $\|P^{\odot 2}v\|^2 = \frac{1}{m}\sum_\ell f(X_\ell)^2$. We have

$$\check{F}_c - \mathbb{E}[\check{F}_c] = \frac{1}{m}\sum_\ell (\check{f}(X_\ell) - \bar{f})^2 - \mathbb{E}[\check{f}(X_\ell - \bar{f})^2]$$

$$= \check{F} - \mathbb{E}[\check{F}] + \frac{1}{m}\sum_\ell -2\check{f}(X_\ell)\bar{f} + \bar{f}^2 + 2\mathbb{E}[\check{f}(X_\ell)\bar{f}] - \bar{f}^2$$

$$= \check{F} - \mathbb{E}[\check{F}] + 2\bar{f}^2 - \frac{2}{m}\bar{f}\sum_\ell \check{f}(X_\ell).$$

Using (3) and $|f(X)| \le |E|$, we have

$$|\check{F} - \mathbb{E}[\check{F}]| \le |\check{F}_c - \mathbb{E}[\check{F}_c]| + 4\delta_{\mathcal{E}}|E|^2. \tag{5}$$

We can apply Lemma 22 to $\bar{S}^{\odot 2}v$ and $\check{F}$, as $f$ and $g$ in that lemma, using that $\check{F} = \|\bar{S}^{\odot 2}v\|^2$ with failure probability at most $\delta_{\mathcal{E}}$, and that $\|\bar{S}^{\odot 2}v\|^2 \le |E|^2$. Using $t = \varepsilon\|v\|^2/2 + 4\delta_{\mathcal{E}}|E|^2$ in that lemma, together with (5) and (4), we have

$$\Pr[|\|\bar{S}^{\odot 2}v\|^2 - \|v\|^2| > \varepsilon\|v\|^2/2 + 4\delta_{\mathcal{E}}|E|^2 + \delta_{\mathcal{E}}|E|^2] \le \Pr[|\check{F} - \mathbb{E}[\check{F}]| > \varepsilon\|v\|^2/2 + 4\delta_{\mathcal{E}}|E|^2]$$

$$\le \Pr[|\check{F}_c - \mathbb{E}[\check{F}_c]| > \varepsilon\|v\|^2/2] + \delta_{\mathcal{E}}$$

$$\le \delta/2 + \delta_{\mathcal{E}} \le \delta.$$

The theorem follows, using $5\delta_{\mathcal{E}}|E|^2 = 5\delta^3/4m^2 < \varepsilon\|v\|^2/2$, possibly adjusting $m$ by a constant factor. $\qquad\square$

### B.3.2 Bundles of Bindings of Multiple Vectors

We can again hope to use Corollary 23 in this setting. However, to obtain the appropriate constants $c_1, \ldots, c_n$ to use in Corollary 23, we cannot directly apply Theorem 25; it is only a base case, for when $k = 2$. We proceed via induction on the maximum binding size.

**Lemma 32.** *Given hypergraph $G = (V, E)$, let $f(x_1, \ldots, x_{|V|}) = \sum_{e\in E}\prod_{i\in e} x_i$, and let $k$ be the maximum hyperedge size. Let $X = (X_1, \ldots, X_{|V|})$ be a tuple of independent random Rademacher variables, i.e., each takes values $\{\pm 1\}$ with probability $\frac{1}{2}$. For $\delta_0 \le O\left(1/\sqrt{|E|}\right)$:*

$$\Pr\left[|f(X)| \ge 2\sqrt{k!\cdot|E|}\log^{k/2}(k/\delta_0)\right] \le \delta_0$$

*Proof.* We proceed by induction on $k$. When $k = 2$, we are done by Theorem 11.

To handle general $k$, we again apply the version of McDiarmid in Corollary 23. If we fix $(x_1, \ldots, x_{|V|})$, changing the value of $x_i$ again changes $f$ by at most $\sum_{e\in E: i\in E}\prod_{j\in E: j\ne i} x_j$. The expression $\sum_{e\in E: i\in E}\prod_{j\in E: j\ne i} x_j$ encodes a hypergraph $G_i = (V_i, E_i)$ with maximum edge size $k - 1$, so by induction, we may assume:

$$\Pr\left[\left|\sum_{e\in E: i\in E}\prod_{j\in E: j\ne i} x_j\right| \ge 2\sqrt{(k-1)!|E_i|}\log^{(k-1)/2}(k/\delta_0)\right] \le \frac{(k-1)\delta_0}{k}$$

Now, we invoke Corollary 23 with $c_i = 2\sqrt{(k-1)!|E_i|}\log^{(k-1)/2}(k/\delta_0)$.

$$\sum_{i=1}^{|V|} c_i^2 = \sum_{i=1}^{|V|} 4(k-1)!|E_i|\log^{k-1}(k/\delta_0)$$

$$= 4(k-1)!\log^{k-1}(k/\delta_0)\sum_{i=1}^{|V|}|E_i|$$

$$\leq 4(k-1)!(\log^{k-1}(k/\delta_0)) \cdot k|E|$$

$$= 4k! \cdot |E|\log^{k-1}(k/\delta_0)$$

We can also set $t = \sqrt{2}\sqrt{k! \cdot |E|}\log^{k/2}(k/\delta_0)$. Then,

$$t^2 = 2k! \cdot |E|\log^k(k/\delta_0) \geq \frac{1}{2}\log(k/\delta_0)\sum_{i=1}^{|V|} c_i^2$$

Finally, setting $\delta = \frac{(k-1)\delta_0}{k}$, $M = |E|$, Corollary 23 implies:

$$\Pr\left[|f(X)| \geq \sqrt{2}\sqrt{k! \cdot |E|}\log^{k/2}(k/\delta_0) + \frac{(k-1) \cdot |E| \cdot \delta_0}{k}\right] \leq \frac{\delta_0}{k} + \frac{(k-1)\delta_0}{k} \leq \delta_0$$

To complete the proof, it suffices to show

$$\frac{(k-1) \cdot |E| \cdot \delta_0}{k} \leq (2-\sqrt{2})\sqrt{k! \cdot |E|}\log^{k/2}(k/\delta_0)$$

However, given our upper bound on $\delta_0$, this holds for all values of $k$. □

**Corollary** (Restatement of Corollary 12). *For scaled sign $\bar{S}^{\odot k} \in \{\pm\frac{1}{\sqrt{m}}\}^{\binom{d}{k}}$, $v \in \{0,1\}^{\binom{d}{k}}$, and $i \in [m]$, there is $C > 0$ and $m = O(\varepsilon^{-2}C^{k\log k}\log^{k+1}(k\|v\|_1/(\varepsilon\delta)))$ such that $\Pr[|\|\bar{S}^{\odot k}v\|^2 - \|v\|^2| > \varepsilon\|v\|^2] \leq \delta$.*

Just as there was a graphical view of $\bar{S}^{\odot 2}v$, we have a natural way to view $\bar{S}^{\odot k}v$ as a hypergraph $H = (V, E)$. For each $\ell \in [m]$, let $X_\ell = (x_1, x_2, \ldots, x_{|V|})$ be the Rademacher variables contributing to the summands of $\bar{S}_\ell^{\odot k}, *v$, the $\ell$-th entry of $\bar{S}^{\odot k}v$, and we can write

$$f(X_\ell) \equiv \sqrt{m}\bar{S}_\ell^{\odot k} = \sum_{(i_1,\ldots,i_k)\in E} x_{i_1} x_{i_2} \cdots x_{i_k}$$

*Proof.* We will follow the same approach to the proof of Theorem 11. Due to the similarity of the proofs, we will omit steps here, but we will keep track of the parameters and calculations that differ.

Fix $i \in [|V|]$, and define $g_i(x_1, \ldots, x_{|V|}) = x_i\sum_{e\in E:i\in e}\prod_{j\in e,j\neq i} x_j$. For each coordinate $\ell \in [m]$, flipping the sign of $x_i$ will change $f(X_\ell)$ by at most $\sum_{e\in E:i\in e}\prod_{j\in e,j\neq i} x_j$. Lemma 32 gives us a high probability bound on the magnitude of $\sum_{e\in E:i\in e}\prod_{j\in e,j\neq i} x_j$:

$$\Pr\left[2|g_i(X_\ell)| > 2\sqrt{(k-1)! \cdot 3^{k-1} \cdot \deg_G(i) \cdot \log^{(k-1)}(k|E|m/\delta)}\right] \leq \frac{\delta^3}{4k|E|^3m^3}$$

Choosing this upper bound on the probability ensures that we obtain the same $\delta_{\mathcal{E}}$ as we do in Theorem 11. Defining $\check{f}(X_\ell)$ as before, Theorem 21 says

$$\Pr\left[|\check{f}(X_\ell) - \mathbb{E}[\check{f}(X_\ell)]| \geq t\right] \leq 2\exp\left(-\frac{t^2}{k! \cdot 3^{k-1} \cdot |E|\log^{(k-1)}(k|E|m/\delta)}\right)$$

The proxy variance of $\check{f}(X_\ell) - \mathbb{E}[\check{f}(X_\ell)]$ is now

$$\sigma_{\check{f}}^2 \equiv k! \cdot 3^{k-1} \cdot |E|\log^{(k-1)}(k|E|m/\delta)$$

Defining $\check{F}$ and $\check{F}_c$ the same way, and choosing large enough $m = O(\varepsilon^{-2} \cdot k!^2 \cdot 3^{2k} \cdot \log^{k+1}(k|E|/(\varepsilon\delta)))$, we have

$$\Pr\left[|\check{F}_c - \mathbb{E}[\check{F}_c]| > \varepsilon|E|/2\right] \le \delta/2$$

The remainder of the proof follows without changes after we fix the value of $m$. Regarding the dependence on $k$,

$$\log(k!^2 3^{2k}) \le \log(2\pi k(k/e)^{2k} e^{1/6k} 3^{2k}) = 2k\log(3k/e) + O(\log k) = O(k\log k),$$

using an inequality form of Stirling's approximation. The corollary follows. $\qquad\square$

# C AUTOCORRELATION ASSOCIATIVE MEMORIES AS BUNDLES OF ROBUST BINDINGS

As discussed in §3.2, this section provides a new analysis of Hopfield networks, and proposes a Hopfield variant that is a space-efficient VSA bundling operation.

## C.1 ANALYSIS OF HOPFIELD NETWORKS VIA CONCENTRATION BOUNDS

Our analysis is akin to that of McEliece et al. (1987), but uses concentration bounds instead of large deviation bounds holding in the limit. The results are similar, in leading terms.

**Theorem** (Restatement of Theorem 13). *Given matrix $S \in \{\pm1\}^{m \times n}$ with uniform independent entries, $j \in [n]$, and $\delta \in (0,1]$. If $y \in \{0,\pm1\}^m$ with $y^\top S_{*j}/\|y\| \ge 2\sqrt{n\log(2m/\delta)}$, then with failure probability at most $\delta$, $\text{sign}_{\ge}((SS^\top - nI_m)y) = S_{*j}$. Here it is assumed that the coordinates $i$ at which $y_i \ne S_{ij}$ are chosen before $S$, or without knowledge of it.*

Since $m \ge (y^\top S_{*j})^2/\|y\|_1$, a necessary condition here is that $m \ge 4n\log(2m/\delta)$. When $y = S_{*j}$, $y^\top S_{*j} = m = \|y\|_1$, and $m \ge 4n\log(2m/\delta)$ suffices. There is

$$m = (1 + o(1))4n\log(2n/\delta) \text{ as } n/\delta \to \infty$$

such that $m \ge 4n\log(2m/\delta)$.

*Proof.* Let $I_{-j} \in \mathbb{R}^{n \times n}$ denote the identity matrix with its $j$'th diagonal entry set to zero. We have

$$(SS^\top - nI_m)y = S_{*j}S_{*j}^\top y - y + (SI_{-j}S^\top - (n-1)I)y.$$

The first coordinate of $(SI_{-j}S^\top - (n-1)I)y$ is $\sum_{\substack{j' \in [n] \\ j' \ne j}} \sum_{1 < j'' \le m} S_{1j'}S_{j'j''}y_{j''}$, which is a sum of at most $(n-1)(\|y\|_1 - 1) \le n\|y\|_1$ independent $\pm1$ values. By Theorem 25, with failure probability at most $2\exp(-\alpha)$, this sum is bounded by $\sqrt{2\alpha n\|y\|_1}$ in magnitude. For $\alpha = \log(2m/\delta)$, with failure probability $\delta$, every coordinate of $(SI_{-j}S^\top - (n-1)I)y$ is at most $\sqrt{2\alpha n\|y\|_1}$.

Assuming this bound, if $S_{*j}^\top y - 1 > \sqrt{2\alpha n\|y\|_1}$, then $S_{ij}S_{*j}^\top y - y_i$ agree in sign with $S_{ij}$, and will exceed $((SI_{-j}S^\top - (n-1)I)y)_i$ in magnitude, for all $i \in [m]$, and therefore $\text{sign}_{\ge}(SS^\top y) = S_{*j}$. For this it is enough if $S_{*j}^\top y/\sqrt{\|y\|_1} > 2\sqrt{\alpha n} = 2\sqrt{n\log(2m/\delta)}$, as claimed. $\qquad\square$

**Relation to VSA: bundling of bindings** Suppose $m$ is even, and we can write $S_{*j} = \left[\begin{smallmatrix} x \\ y \end{smallmatrix}\right]$, for $x, y \in \{\pm1\}^{m/2}$. Then assuming the above conditions, there is $m = O(n\log(n/\delta))$ so that the Hopfield output on input $\left[\begin{smallmatrix} x \\ 0 \end{smallmatrix}\right]$ or $\left[\begin{smallmatrix} 0 \\ y \end{smallmatrix}\right]$ will be $S_{*j}$. In this special case, the Hopfield network does a bit more than a bundle of pairwise bindings can do: it returns the other vector of a binding, without performing membership tests over a dictionary of vectors (cleanup). This should not be entirely surprising, since $\left[\begin{smallmatrix} x \\ y \end{smallmatrix}\right]\left[\begin{smallmatrix} x \\ y \end{smallmatrix}\right]^\top = \left[\begin{smallmatrix} xx^\top & xy^\top \\ yx^\top & yy^\top \end{smallmatrix}\right]$, and the diagonal entries of $xy^\top$ and $yx^\top$ form the Hadamard product vector (i.e. element-wise product vector) $x \circ y$, which is the MAP-I binding of $x$ and $y$.

Looking at the weight matrix $W = \sum_{j=1}^n S_{*j}S_{*j}^\top$, the two particular diagonals observed above contain exactly the MAP-I bundling of pairwise bindings of MAP-I atomic vectors, of dimension

equal to half the net size. Also, the outer product $xy^\top$ is the binding operator for $x$ and $y$ proposed by Smolensky (1990), and the relation of this binding operator to the MAP-I binding as a reduced representation is well-known via Frady et al. (2021).

As mentioned in the introduction, much of this discussion, including the weight matrix as a sum of outer products of the input vectors, the fixed-point condition for representation, the mapping to neural networks, and reconstruction from erasures, goes back at least to Nakano (1972); Kohonen (1972); Anderson (1972).

**Thinning the weight matrix**  By Theorem 13, it suffices if $\|y\|_1 \geq 2n\log(2m/\delta)$ for the theorem's conclusion to apply, so that the $S_{*j}$ matching $y$ can be reconstructed. That is, for diagonal matrix $V \in \{0,1\}^{m \times m}$, if $\|Vy\|_1 \geq 2n\log(2m/\delta)$ is large enough, $\text{sign}_{\geq}(SS^\top Vy) = S_{*j}$. We could regard this, for given $V, y$, as $\text{sign}_{\geq}(Wy)$ for a matrix $W \equiv SS^\top V$ with $\|V\|_F^2$ nonzero columns: we erase columns of $SS^\top$, not entries of $y$. That is, if $n$ vectors are to be stored, but the vector dimension is fixed at larger than $m \gg 2n\log(2m/\delta)$, the necessary number of stored entries is smaller than $m^2$.

**VSAs, Autocorrelation Associative Memories, and modern Hopfield nets.**  A different neural network model was proposed in recent years by Krotov & Hopfield (2016), sometimes called a *modern* Hopfield network, which also uses a dynamic process to produce its output. Here the output of an iteration is more "peaked;" for example, a version by Demircigil et al. (2017) uses softmax when computing the next iterate. This model has a higher capacity than classical associative memories using sums of outer products. However, it stores all its input vectors explicitly, so that the main question is not whether the given vectors are *represented* (since they are stored), but under what circumstances the dynamic process *converges*. The number of entries stored in this model is $\Theta(mn)$, where (classic) associative networks store $\Theta(m^2)$, and VSAs store $\Theta(m)$. The different amounts of storage imply different capabilities:

- The modern Hopfield model gives a high-capacity way to do maximum inner product search on a set of vectors, using a neural network formalism.
- The associatron/original Hopfield model supports the reconstruction of a vector in a stored set of vectors from an erased or noisy version.
- VSAs support membership tests, telling us whether two given vectors are present as a bound pair in a set.

### C.2  HOPFIELD±: AUTOCORRELATION ASSOCIATIVE MEMORIES AS VSAS

We can also use an AAM/Hopfield model (a sum of vector outer products) to do set intersection estimation, as in VSAs. Here we represent a bundle as the sum of outer products of a vector with itself, as in Smolensky (1990). The key theorem, proven below, is the following.

**Theorem** (Restatement of Theorem 14).  *Given $\varepsilon, \delta \in (0, 1]$, scaled sign matrix $\bar{S} \in \frac{1}{\sqrt{m}}\{\pm 1\}^{m \times d}$ with uniform independent entries, diagonal matrix $V \in \mathbb{R}^{d \times d}$, and diagonal matrix $D \in \{0, \pm 1\}^{d \times d}$ with uniform independent $\pm 1$ diagonal entries. There is $m = O(\varepsilon^{-1}\log(d/\delta)^2)$ such that with failure probability $\delta$, $\|\bar{S}VD\bar{S}^\top\|_F^2 = (1 \pm \varepsilon)\|V\|_F^2$.*

Since $\bar{S}$ is a JL projection matrix, it will, with high probability, preserve the norms of the columns of $V$, and multiplying by $\bar{S}^\top$ on the right will preserve the norms of the rows of $\bar{S}V$. We can think of this as $\bar{S}$ satisfying the JL property with respect to the operation $\bar{S}VD\bar{S}^\top$. The novelty here is that the dependence of $m$ on $\varepsilon$ is $\varepsilon^{-1}$, not $\varepsilon^{-2}$; on the other hand, $\bar{S}VD\bar{S}^\top$ comprises $m^2$ values, not $m$. This is yet another route to preserving the norm of a vector, stored as the diagonal of $V$, using the same storage and speed of computation (up to log factors) as that of simply using a larger projection matrix on one side, as in MAP-I. In other words, we obtain a bundling operator with performance characteristics similar to those of MAP-I, but using fewer random bits.

This theorem is analogous to Lemma 2. The following is analogous to Corollary 4, and can be proven in exactly the same way. It implies in particular that for diagonal matrices $X, Y$ representing sets, with 0/1 diagonal entries encoding element membership, the size of their intersection can be estimated accurately using their "Hopfield net" representations.

We have a direct analog of Corollary 4, stated below. Analogs of Corollary 3 and Theorem 5 also follow directly; we omit the proofs.

**Corollary 33.** *Under the conditions of Theorem 14 (the JL property for $\bar{S}VD\bar{S}^\top$), for diagonal $X, Y \in \mathbb{R}^{d \times d}$, there is $m = O(\varepsilon^{-2} \log(d/\delta)^2)$ so that $\mathrm{tr}((\bar{S}XD\bar{S}^\top)(\bar{S}YD\bar{S}^\top)) = \mathrm{tr}(XY) \pm \epsilon \|X\|_F \|Y\|_F$ with failure probability $\delta$.*

Note that $\mathrm{tr}(XY)$ is here the Frobenius product of $X$ and $Y$. We omit the proof; it is isomorphic to the proof of Corollary 4.

To prove Theorem 14, we will use the Hanson-Wright inequality. In its statement,

$$\|X\|_{\psi_2} = \inf\{K \mid \mathbb{E}(\exp(X^2/K^2) \le 2\}$$

is the sub-Gaussian norm of random variable $X$, which in our case, for a Rademacher variable, will be $1/\sqrt{\log 2}$.

**Theorem 34** (Hanson-Wright inequality, Rudelson & Vershynin (2013)). *Let $x \in \mathbb{R}^d$ be a random vector with independent entries, $\mathbb{E}[x] = 0$, and $\|x_i\|_{\psi_2} \le K$ for some positive constant $K$. Let $A$ be an $n \times n$ matrix. Then for every $t \ge 0$,*

$$\Pr\left\{ |x^\top A x - \mathbb{E}[x^\top A x]| > t \right\} \le 2 \exp\left[ -c \min\left( \frac{t^2}{K^4 \|A\|_F^2}, \frac{t}{K^2 \|A\|} \right) \right],$$

*where $c$ is some positive constant. In particular, $t = \frac{K^2}{c} \|A\|_F \log(2/\delta)$ yields a failure probability bound of at most $\delta$, when $\log(1/\delta) \ge c$.*

*Proof of Theorem 14.* Let $E = \bar{S}^\top \bar{S} - I$; since the diagonal entries of $E$ are zero, we have

$$\begin{aligned}
\|\bar{S}VD\bar{S}^\top\|_F^2 = \mathrm{tr}(\bar{S}VD\bar{S}^\top \bar{S}VD\bar{S}^\top) &= \mathrm{tr}(VD(I+E)VD(I+E)) \\
&= \mathrm{tr}(V^2 D^2) + 2\,\mathrm{tr}(V^2 D^2 E) + \mathrm{tr}(VDEVDE) \\
&= \|V\|_F^2 + 0 + \mathrm{tr}(VDEVDE) \\
&= \|V\|_F^2 + 0 + \sum_{i \in [d]} (VDE)_{i*}(VDE)_{*i} \\
&= \|V\|_F^2 + 0 + \sum_{i,j \in [d]} V_{ii} D_{ii} E_{ij} V_{jj} D_{jj} E_{ji} \\
&= \|V\|_F^2 + b^\top V(E \circ E) V b,
\end{aligned} \qquad (6)$$

where in the last step $b \in \{\pm 1\}^d$ comprises the diagonal entries of $D$, and we use the symmetry of $E$. The simplifications also use the fact that $V$ and $D$ commute and $D^2$ is a projection matrix.

We will apply the Hanson-Wright inequality, for which we need to bound $\|V(E \circ E)V\|_F^2$. First, we bound the entries of $E$: each such entry is $\frac{1}{m}$ times a sum of $m$ independent Rademacher values, and therefore from Theorem 25 is at most $\frac{1}{\sqrt{m}} \sqrt{2 \log(2/\delta_1)}$ with failure probability $\delta_1$. By applying a union bound to all $d(d-1)$ off-diagonal entries of $E$, we have that all such entries are at most $\frac{2}{\sqrt{m}} \sqrt{\log(2d/\delta_1)}$ with failure probability $\delta_1$. Assuming this event, the square of each entry is at most $\frac{4}{m} \log(2d/\delta_1)$. We have

$$\|V(E \circ E)V\|_F^2 = \sum_{i,j \in [d]} V_{ii}^2 V_{jj}^2 E_{ij}^4 \le \frac{16 \log(2d/\delta_1)^2}{m^2} \sum_{i,j \in [d]} V_{ii}^2 V_{jj}^2 = \frac{16 \log(2d/\delta_1)^2}{m^2} \cdot \|V\|_F^4.$$

We will apply Theorem 34 with $x$ and $A$ of the theorem mapping to $b$ and $V(E \circ E)V$. We have $\mathbb{E}_b[b^\top V(E \circ E)V b] = 0$, since off-diagonal terms with $\mathbb{E}_b[b_i b_j \ldots], i \neq j$ are zero, and the diagonal entries of $E$ are zero. The value of $t$ in the last line of Theorem 34 translates here to

$$b^\top V(E \circ E)V b \le \frac{c}{\log(2)} \cdot \frac{4 \log(2d/\delta_1)}{m} \cdot \|V\|_F^2 \log(1/\delta_2)$$

with failure probability $\delta_2$, using $\|X\|_{\psi_2} = 1/\sqrt{2}$, for $X \sim \{\pm 1\}$. Putting this together with (6), using a union bound for the events with failure probabilities $\delta_1$ and $\delta_2$, and reducing the precise bound above to $O()$ notation, yields the result. $\qquad \square$

## D  ANALYSIS OF MAP-B

We will analyze membership testing for MAP-B, for bundles (§D.1), sequences of sets as encoded with rotations (§D.2), and bundles of bound key-value pairs (§D.3). Here being a key-value pair means that some restrictions are placed on members of a bound pair, in particular that the keys come from a set that is disjoint from the set of values, and so there is greater independence among the relevant random variables.

Recall that the MAP-B bundling operator is $\text{sign}(Sv)$, where $S$ is a sign matrix, Def. 1, and the sign function takes a random $\pm 1$ value when its argument is zero.

### D.1  BUNDLING

#### D.1.1  MEMBERSHIP TEST

We first consider testing membership in a MAP-B bundling of $n$ atomic vectors.

**Theorem** (Restatement of Theorem 15). *For $v \in \{0,1\}^d$, let $x = \text{sign}(Sv)$ be the MAP-B bundling of $n = \|v\|_1$ atomic vectors. Then for all $i \in [m]$ and $j \in \text{supp}(v)$, $\Pr[x_i S_{ij} = +1] = 1/2 + \Theta(1/\sqrt{n})$, as $n \to \infty$, and there is $m = O(n \log(d/\delta))$ such that with failure probability $\delta$, $j \in [d]$ has $j \in \text{supp}(v)$ if and only if $x^\top S_{*j} = e_j^\top S \, \text{sign}(Sv)$ has $x^\top S_{*j} \geq \sqrt{2m \log(2d/\delta)}$.*

Note that MAP-B bundling is not associative, so analysis of this bundling operation is not the end of the story, as discussed in §D.1.2.

*Proof.* We consider $S_{*j}^\top x$ in the two cases where $j$ is, and is not, in $\text{supp}(v)$. Fix some $\hat{j} \in \text{supp}(v)$, and let $a = S_{1\hat{j}}$ and $b = (Sv)_1 - a$, so that $x_1 = \text{sign}(a + b)$, and $a \, \text{sign}((a+b)) = \text{sign}(a(a+b))$ is a summand of $S_{*\hat{j}}^\top x$. We consider the cases where $n - 1$ is even vs. odd. Let $\mathcal{E}$ denote the event $\text{sign}(a(a+b)) > 0$.

Since the $n - 1$ summands of $b = (Sv)_1 - a$ are i.i.d. Rademacher variables, the probability that $b = 0$ is, for $n - 1$ even, equal to $\binom{2p}{p} \frac{1}{2^{2p}} = \frac{1}{\sqrt{\pi p}}(1 + O(1/p)) = \sqrt{\frac{2}{\pi n}}(1 + O(1/n))$, where $p = (n-1)/2$, using Stirling's approximation, or the related expression for the Catalan numbers. For $n - 1$ even, when $b \neq 0$, $|b| >= 2$, so $a + b > 0$, and $\Pr[\mathcal{E} | b \neq 0] = 1/2$. We have

$$\Pr[\mathcal{E}] = \Pr[\mathcal{E} \mid b = 0] \Pr[b = 0] + \Pr[\mathcal{E} \mid b \neq 0] \Pr[b \neq 0]$$

$$= \Pr[b = 0] + \frac{1}{2}(1 - \Pr[b = 0])$$

$$= \frac{1}{2} + \frac{1}{2}\sqrt{\frac{2}{\pi n}}(1 + O(1/n)) = \frac{1}{2} + \sqrt{\frac{1}{2\pi n}}(1 + O(1/n)).$$

When $n - 1$ is odd, $b$ cannot be zero, while

$$\Pr[b = 1] = \Pr[b = -1] = \binom{n-1}{n/2 - 1}\frac{1}{2^{n-1}} = \frac{n/2}{n}\binom{n}{n/2}\frac{2}{2^n} = \frac{1}{\sqrt{\pi(n/2)}}(1 + O(1/n)).$$

Considering the rounding for the case $a + b = 0$, we have $\Pr[\mathcal{E} \mid b = \pm 1] = 3/2$. Therefore,

$$\Pr[\mathcal{E}] = \frac{3}{2}\Pr[b = \pm 1] + \frac{1}{2}\Pr[b \neq \pm 1]$$

$$= \frac{1}{2} + \Pr[b = \pm 1]$$

$$= \frac{1}{2} + \sqrt{\frac{2}{\pi n}}(1 + O(1/n)).$$

So

$$\Pr[x_1^{(1)}x_1 = 1] = \Pr[\text{sign}(a(a+b)) > 0]$$

$$= \Pr[\mathcal{E}] \geq 1/2 + \sqrt{1/2\pi n}(1 + O(1/n))$$

$$> 1/\sqrt{7n} \text{ for large enough } n.$$

This applies to all coordinates of all vectors $S_{i,j}$ for $j \in \text{supp}(v)$, and inspection of the expressions shows the first claim of the theorem, $\Pr[x_i S_{ij} = +1] = 1/2 + \Theta(1/\sqrt{n})$ for such $i, j$.

For the remaining claim: we have that $x^\top S_{*j}$ for $j \in \text{supp}(v)$ is a sum of $\pm 1$ independent values, with $+1$ having probability $1/2 + \Theta(1/\sqrt{n})$, and at least $1/2 + \sqrt{1/7n}$, for large enough $n$; this implies expectation at least $m/\sqrt{7n}$. Hoeffding's inequality implies that $\Pr[x^\top S_{*j} - m/\sqrt{7n} < -\sqrt{2m\log(2d/\delta)}] \leq \delta/2d$. Now suppose $j \notin \text{supp}(v)$. Then using Hoeffding again, we obtain:

$$\Pr[x^\top S_{*j} > \sqrt{2m\log(2d/\delta)}] \leq \delta/2d.$$

Thus, there is $m = O(n\log(d/\delta))$ such that $m/\sqrt{7n} > 2\sqrt{2m\log(2d/\delta)}$, and therefore by a union bound for all $j \in [d]$, with failure probability at most $\delta$, $x^\top S_{*j} > \sqrt{2m\log(2d/\delta)}$ if an only if $j \in \text{supp}(v)$. $\qquad\square$

### D.1.2 DEPENDENCE ON DEPTH

Instead of bundling a set of elements all at once, suppose instead that bundling is done as a sequence of operations. We will see that for some such sequences, with high depth, so much information is lost that accurate membership tests become impossible.

**Lemma** (Restatement of Lemma 16). *Given independent sign vectors $x^{(i)} \in \{\pm 1\}^m, i \in [r]$, construct vector $x$ by setting $x \leftarrow x^{(1)}$, and for $j = 2, \ldots, r$, setting $x \leftarrow sign(x + x^{(j)})$. Then, for $\ell \in [m]$,*

$$\Pr[x_\ell^{(1)} x_\ell = 1] = 1/2 + 1/2^r$$

*Proof.* We consider $x_1^{(1)} x_1$; the same analysis applies to $x_\ell^{(1)} x_\ell$ for $\ell > 1$. Let $a$ denote $x_1^{(1)}$, and let $z_{(j)}$ denote the value of $x_1$ just after the assignment $x \leftarrow \text{sign}(x + x^{(j)})$ in computing $x$. Since $b\,\text{sign}(c + e) = \text{sign}(bc + be)$ for $b = \pm 1$, we have that

$$x_1 x_1^{(1)} = az_{(r)} = \text{sign}(az_{(r-1)}) + ax^{(r)}).$$

By induction on $j$ with this base case, $az_{(r-j)} = \text{sign}(az_{(r-(j+1))}) + ax^{(r-j)})$, ending with $az_{(1)} = a^2 = 1$. Suppose inductively on $j$ (in the other direction) that $az_{(j)}$ is $+1$ with probability $1/2 + 1/2^j$, respectively; this is true for $j = 1$, as noted. We have $az_{(j)} = \text{sign}(az_{(j-1)}) + ax^{(j)})$, and $ax^{(j)} = \pm 1$ with equal probability independent of $az_{(j-1)}$. Analysis of the four combinations $az_{(j-1)}) = \pm 1, ax^{(j)} = \pm 1$ and their probabilities shows that the inductive step holds, that is, $az_{(j+1)}$ is $+1$ with probability $1/2 + 1/2^{j+1}$. In particular, $\Pr[x_1 x_1^{(1)} = az_{(r)} = +1] = 1/2 + 1/2^r$, as claimed. $\qquad\square$

Note that this property of $x_\ell x_\ell^{(1)}$ does not require the $x^{(i)}$ to be atomic vectors of the VSA. Considering such summands for $x^\top x^{(1)}$ leads to $\mathbb{E}[x^\top x^{(1)}] \approx m/2^r$, as compared to at least $m/\sqrt{7n}$ as in Theorem 15, with a corresponding change in $m$ needed to obtain effective membership testing. Suppose all the $x^{(i)}$ are bundles, and $x$ is the root of a tree of bundlings of a total of $n$ vectors. Then $x^{(1)}$ is a leaf of a tree of bundling operations of depth at least $r$, with possibly $n = r$, when all $x^{(i)}$ involved with $x^{(1)}$ are atomic; $n = 2^r$, when $x$ is the root of a complete binary tree of depth $r$; or $n > 2^r$, when $x^{(1)}$ is not at the maximum depth, or when a single bundling operation involves more than two vectors. Thus for membership testing to be effective, $m$ might need to be at large as $4^n$, or as small as $\tilde{O}(n)$, depending on operation depth in computing $x$.

The following lemma puts $x^{(1)}$ in a more general bundle.

**Corollary 35.** *Given independent sign vectors $x^{(i)} \in \{\pm 1\}^m$ for $i \in [n']$ and $y^{(j)} \in \{\pm 1\}^m$ for $j \in [r]$, construct vector $x$ by setting $x \leftarrow sign(\sum_{i \in n'} x^{(i)})$, and then for $j = 1, \ldots, r$, setting $x \leftarrow sign(x + y^{(j)})$. Then, for $\ell \in [m]$,*

$$\Pr[x_\ell^{(1)} x_\ell = 1] = \frac{1}{2} + \frac{1}{2^r} \Theta\left(\frac{1}{\sqrt{n}}\right)$$

*Proof.* Combine the analysis of Theorem 15 with the lemma just above. $\qquad\square$

Note that this corollary covers any sequence of operations yielding a vector of the form $x^{(1)} \oplus y$, that is $x^{(1)}$ bundled with $y$, regardless of how $y$ was computed in the MAP-B algebra.

### D.1.3 Testing for Empty Intersection

In this section, rather than estimating set intersection size, we aim for bounds on $m$ so that we can distinguish between nonempty and empty set intersections. Estimating the set intersection *size* will depend on a rather complicated convolution of Rademacher random variable; the question of distinguishing emptiness has a much cleaner analysis.

Throughout this section, we assume that the elements comprising $X$ and $Y$ were all bundled during the same operation. In other words, they correspond to a depth-1 bundling tree, in the language of the previous section.

**Lemma 36.** *Let $X, Y \subseteq [d]$, and use $x$ and $y$ to denote their respective MAP-B vectors. If we choose dimension $m \geq \Omega\left(\log(1/\delta) \cdot |X||Y|\right)$, we can distinguish between the cases $|X \cap Y| = 0$ and $|X \cap Y| \geq 1$, using the criterion of whether $x^T y$ is smaller or greater than $\sqrt{2m \log(2/\delta)}$.*

*Proof.* Let $v, w \in \{0,1\}^d$ be characteristic vectors for the sets $X, Y \subseteq [d]$. First, if $|X \cap Y| = 0$, their MAP-B bundles $x = \text{sign}(Sv)$ and $y = \text{sign}(Sw)$ are independent, and for any coordinate $\ell \in [m]$:
$$\mathbb{E}[x_\ell y_\ell] = \Pr[x_\ell y_\ell = 1] - \Pr[x_\ell y_\ell = -1] = 0$$

Using Theorem 25, we conclude that with probability $1 - \frac{\delta}{2}$, we satisfy $x^\top y \leq \sqrt{2\log(2/\delta) \cdot m}$.

Now, suppose $|X \cap Y| = 1$. Then, we can use $e^*$ to denote the standard basis vector that represents the element in their intersection. Let $x = \text{sign}(Sv)$ and $y = \text{sign}(Sw)$ be the MAP-B bundles representing $X$ and $Y$. (Here, $S$ is a random sign matrix; see Definition 1.)

Since all coordinates within $x$ are independent of each other (and likewise for $y$), it suffices to understand the distribution of a single coordinate. Fix $\ell \in [m]$. We will study the quantities $\Pr[x_\ell y_\ell = 1]$ and $\Pr[x_\ell y_\ell = -1]$ to eventually obtain a concentration bound for $x^\top y$.

Let $Z^{(X)} = (Se^*)_\ell + \sum_{i \in X \setminus Y} Z_i^{(X)}$, where the $\{Z_i^{(X)}\}_{i=1}^m$ are independent Rademacher variables coming from the $\ell$-th coordinates of the atomic vectors that were bundled to form $X$. Define $Z^{(Y)}$ similarly for $Y$, and let $X' = Z^{(X)} - (Se^*)_\ell$ and $Y' = Z^{(Y)} - (Se^*)_\ell$.

First, assume $|X \setminus Y|$ and $|Y \setminus X|$ are both even. The other cases follow similarly; the only change to track is that when $|X \setminus Y|$ (or $|Y \setminus X|$) is odd, $Z^{(X)}$ (resp. $Z^{(Y)}$) could be 0, and we have a $\frac{1}{2}$ chance of $Z^{(X)}$ (resp. $Z^{(Y)}$) being positive and a $\frac{1}{2}$ chance it is negative.

$$\begin{aligned}
\Pr[x_\ell = y_\ell = 1] &= \Pr[Z^{(X)} = Z^{(Y)} = 1] \\
&= \Pr[X' \geq 0, Y' \geq 0] \cdot \Pr[(Se^*)_\ell = 1] + \Pr[X' \leq 1, Y' \leq 1] \cdot \Pr[(Se^*)_\ell = -1] \\
&= \Pr[X' \geq 0] \cdot \Pr[Y' \geq 0]
\end{aligned}$$

We obtain the following equations similarly, again when $|X \setminus Y|$ and $|Y \setminus X|$ are even.

$$\begin{aligned}
\Pr[x_\ell = 1, y_\ell = -1] &= \Pr[X' \geq 0] \cdot \Pr[Y' < 0] \\
\Pr[x_\ell = -1, y_\ell = 1] &= \Pr[X' < 0] \cdot \Pr[Y' \geq 0] \\
\Pr[x_\ell = y_\ell = -1] &= \Pr[X' < 0] \cdot \Pr[Y' < 0]
\end{aligned}$$

Combining these terms, the quantity $\Pr[x_\ell y_\ell = 1] - \Pr[x_\ell y_\ell = -1]$ is equal to:

$$(\Pr[X' \geq 0] - \Pr[X' < 0]) \cdot (\Pr[Y' \geq 0] - \Pr[Y' < 0]) = \Pr[X' = 0] \cdot \Pr[Y' = 0] \quad (7)$$

Again using the fact that $\binom{2p}{p}\frac{1}{2^{2p}} = \frac{1}{\sqrt{\pi p}}(1 + O(1/p)) = \sqrt{\frac{2}{\pi n}}(1 + O(1/n))$ (which was also used in the membership test), we have $\Pr[x_\ell y_\ell = 1] - \Pr[x_\ell y_\ell = -1] \geq \Omega\left(\frac{1}{\sqrt{|X| \cdot |Y|}}\right)$.

Define $p \equiv C' \cdot \frac{1}{\sqrt{|X| \cdot |Y|}}$, so $\Pr[x_\ell y_\ell = 1] \geq \frac{1+p}{2}$. Applying the Chernoff bound for a sum of $m$ Rademachers, we have:

$$\Pr\left[(x^\top y \geq \frac{m}{2} + \frac{pm}{2} - \sqrt{2\log(2/\delta) \cdot m}\right] \geq 1 - \frac{\delta}{2}$$

In order to distinguish between the two cases, we require $\frac{pm}{2} \geq 2\sqrt{2\log(2/\delta) \cdot m}$, so we need to choose $m \geq \Omega\left(\frac{\log(2/\delta)}{p^2}\right)$. $\qquad\square$

**Remark 37.** *If we have $|X| = n$ and $|Y| = 1$, and enforce a failure probability of $\frac{\delta}{d}$ (so that we can union bound over all atomic vectors that represent $[d]$), we recover the result of Lemma D.1.*

### D.2 ROTATIONS

Our results for MAP-B rotations and bundles of bindings are more limited than what we are able to prove for MAP-I, again due to the nonlinearities present in the binding operation. As noted before, we only consider single element membership testing. For sequences, this means checking if a single element $i$ is present in the $j$-th set in a sequence.

**Theorem 38.** *Given a sign matrix $S \in \mathbb{R}^{m \times d}$ and rotation matrix $R \in \mathbb{R}^{m \times m}$, and integer length $L$. Recall (Def. 7) that $S_{R,L} \equiv [S \ RS \ R^2 S \ \ldots R^{L-1} S]$. For a sequence of $d$-vectors $v_{(0)}, v_{(1)}, \ldots v_{(L-1)} \in \{0, 1\}^d$, let $v \equiv [v_{(0)} \ v_{(1)} \ \ldots v_{(L-1)}]$, let $x = sign(S_{R,L}v)$, and let $n = \|v\|_1$. Then there is $m = O(Ln\log(Ld/\delta))$ such that with failure probability at most $\delta$, $j \in [Ld]$ has $j \in supp(v)$ if and only if $x^\top R^{\lfloor j/d \rfloor} S_{*,j_{\%d}} \geq 2\sqrt{m\log(Ld/\delta)}$, where $j_{\%d} \equiv 1 + (j - 1) \mod d$.*

Here we consider membership testing, leveraging Theorem 15 in a simple way.

*Proof.* Consider

$$x^\top R^{\lfloor j/d \rfloor} S_{*,j_{\%d}} = \sum_{i \in [m]} x_i R^{\lfloor j/d \rfloor} S_{i,j_{\%d}} = \sum_{i \in [m]} S_{i,j_{\%d}} R^{\lfloor j/d \rfloor} (S_{R,L})_{i,*} v$$

$$= 1 + \sum_{i \in [m]} S_{i,j_{\%d}} R^{\lfloor j/d \rfloor} ((S_{R,L})_{i,*} v - S_{i,j_{\%d}}).$$

The random variables of $S$ that appear in $S_{i,j_{\%d}} R^{\lfloor j/d \rfloor} ((S_{R,L})_{i,*} v - S_{i,j_{\%d}})$ cannot appear in

$$S_{i+L,j_{\%d}} R^{\lfloor j/d \rfloor} ((S_{R,L})_{i+L,*} v - S_{i+L,j_{\%d}}),$$

since the appearances of a given $S_{i,j}$ due to shifting in $S_{R,L}$ only span $L$ rows. That is, there are at most $L$ blocks of at least $(m/L) - 1$ rows such that each block can be analyzed as in Theorem 15. We obtain, for a given block of rows, and after taking into account that only $(m/L) - 1$ rows are in a block, that there is $m = O(Ln\log(d/\delta))$ such that with failure probability $\delta$, $j \in [d]$ has $j \in supp(v)$ if and only if the vectors $\tilde{x}$ and $\tilde{S}_{*j}$ corresponding to the block of rows have $\tilde{x}^\top R^{\lfloor j/d \rfloor} \tilde{S}_{*j} \geq 2\sqrt{(m/L)\log(d/\delta)}$. By requiring failure probability at most $\delta/L$, satisfied by $m = O(Ln\log(dL/\delta))$, the result follows. $\qquad\square$

### D.3 BINDING

#### D.3.1 MEMBERSHIP IN KEY-VALUE PAIRS

We will analyze MAP-B bundles of bindings in the restricted setting of key-value pairs. We can test whether a single key $\otimes$ value belongs to such a bundle.

**Definition 39.** *A bundle of key-value pairs is here a bundling of bindings $S^{\odot 2}v$, where $S_{*j}^{\odot 2}$ for $j \in supp(v)$ is $S_{*,q} \circ S_{*,w}$ with $q \in \mathcal{Q}, w \in \mathcal{W}$, where $\mathcal{Q}, \mathcal{W} \subset [d], \mathcal{Q} \cap \mathcal{W} = \{\}$, and a given $q$ appears in only one binding in such a representation. That is, each $S_{*,q}$ is bound to only one $S_{*,w}$, noting that two $q, q'$ may bind to the same $S_{*,w}$.*

**Theorem 40.** *Let $v \in \{0,1\}^{\binom{d}{2}}$ be such that $x = sign(S^{\odot 2}v)$ is a bundle of key-value pairs, as in Def. 39. Let $n = \|v\|_1$. Then there is $m = O(n\log(d/\delta))$ such that with failure probability at most $\delta$, $j \in [\binom{d}{k}]$ has $j \in supp(v)$ if and only if $x^\top S^{\odot 2}_{*j} \geq 2\sqrt{m\log(d/\delta)}$.*

The proof of this theorem again leverages Theorem 15.

*Proof.* For $j \in [\binom{d}{2}]$, let $w(j) \in W, q(j) \in Q$ be the indices such that $S^{\odot 2}_{*,j} = S_{*,w(j)} \circ S_{*,q(j)}$. Let $\hat{j} \in \text{supp}(v)$. We have $\text{sign}(x^\top S^{\odot 2}_{*\hat{j}}) = \text{sign}(1 + (S^{\odot 2}v - S^{\odot 2}_{*,\hat{j}})^\top S^{\odot 2}_{*\hat{j}})$, and

$$S^{\odot 2}v - S^{\odot 2}_{*,\hat{j}} = \sum_{j \neq \hat{j}} S_{*,q(j)} \circ S_{*,w(j)}.$$

Since each $S_{*,q(j)}$ appears only once in the sum, the summands are independent sign vectors. (Whatever the relations among the $w(j)$.) So $x^\top S^{\odot 2}_{*,\hat{j}}$ satisfies the same conditions as does the same expression in the proof of Theorem 15, as does $x^\top S^{\odot 2}_{*,j'}$ for $j' \notin \text{supp}(v)$. Therefore the same conditions on $m$ imply the same results on failure probability as in Theorem 15, and the lemma follows. $\square$

The conclusion of the theorem is much the same as that of Theorem 15. This is not a coincidence: the analysis of key-value pairs reduces to the same setting as that earlier lemma. Binding is done in MAP-B the same as in MAP-I, namely, as coordinate-wise product. Here we also consider membership test in a bundle of bindings, where again we assume that the bundling is done in one step, that is, a sum over the integers followed by the sign function.

# E  SPARSE BINARY BUNDLING AND BLOOM FILTER ANALYSIS VIA CONCENTRATION

We will analyze the VSA dimension needed for reliably estimating the size of the intersection of sets represented by Bloom filters (§E.1) and Counting Bloom filters (§E.2). In the latter, we actually consider a generalization to weighted sets.

## E.1  BLOOM FILTERS

In the VSA model described in this section, the atomic vectors are $x \in \{0,1\}^m$ where for some $k$, $\|x\|_1 \leq k$, with the nonzero entries chosen randomly. As discussed in Def. 17, an atomic vector $B_{*j}$ is created by performing $k$ trials, in each trial picking $i \in [m]$ uniformly at random and setting $B_{ij} \leftarrow 1$. (It is possible to pick the same $i \in [m]$ twice, in which case we only perform the update once.) Bundling will be done with disjunction, so a set with characteristic vector $v \in \{0,1\}^d$ is represented as $1 \wedge Bv$, meaning that the coordinate-wise minimum of $Bv$ with 1 is taken.

This representation, and the resulting membership testing, has long been studied as used as *Bloom filters*, with many variations, including applications to *private* set intersection estimation. However, the narrow question of how to use dot products of Bloom filter representations (or other simple vector operations) to estimate set intersection size, is less well developed. It is outlined briefly in Broder & Mitzenmacher (2004), and discussed in Swamidass & Baldi (2007). An analysis is given in Papapetrou et al. (2010), where the assumption is made that the entries of $Bv$ are independent of each other. [3] We will bound the error in estimating set intersections using Bloom filters in Theorem 18.

A key quantity for our analysis is the number of nonzeros in $Bv$ as a function of $\|v\|_0$, when $v \in \{0,1\}^d$. We will let $\|x\|_0$ denote the number of nonzeros of $x$, so $\|1 \wedge Bv\|_1 = \|Bv\|_0$. This includes real numbers $y$, so $\|y\|_0$ is one when $y \neq 0$, and 0 otherwise.

---

[3]This is a good approximation, but the approximation is not rigorously substantiated, so the results are correct only up to that assumption. See also Bose et al. (2008) regarding difficulties in the analysis of Bloom filters.

**Lemma 41.** *For $v \in \{0,1\}^d$, let $n \equiv \|v\|_0$. Define $\kappa \equiv \|Bv\|_0$, and let $p_\ell \equiv (1 - 1/m)^{k\ell}$. Then $\mathbb{E}[\kappa] = m(1 - p_n)$. Moreover, letting $\tilde{m} \equiv -1/\log(1 - 1/m)$,*

$$1 - \frac{k\ell}{m-1} \le p_\ell = \exp(-k\ell/\tilde{m}) \le \exp(-k\ell/m) \tag{8}$$

*Letting $h_{m,k}(z) \equiv \frac{-\tilde{m}}{k} \log(1 - \frac{z}{m})$, we have $h_{m,k}(\mathbb{E}[\kappa]) = h_{m,k}(m(1 - p_n)) = n$.*

Here, we can think of $h_{m,k}$ as an estimator for $\|v\|_0$, accounting for collisions in $Bv$. It is not unbiased, as we do not have $\mathbb{E}[h_{m,k}(\kappa)] = n$, but $h_{m,k}(\kappa)$ will still be a *good* estimator if $\kappa$ concentrates; luckily for us, it does, as shown below.

The definitions of $\tilde{m}$ may also seem a bit curious; we use the following well-known inequalities.

$$\begin{aligned} \log(1 + x) &\le x \text{ for all } x \\ -\log(1 - x) &\le \frac{x}{1-x} \text{ for } x < 1 \end{aligned} \tag{9}$$

These imply

$$\frac{1}{m} \le -\log(1 - \frac{1}{m}) \le \frac{1/m}{1 - 1/m} = \frac{1}{m-1}, \text{ and so} \tag{10}$$

$$m - 1 \le \tilde{m} \le m \tag{11}$$

*Proof of Lemma 41.* Observe that for each of $n$ vectors $B_{*j}$, there will be $k$ independent trials generating random $i \in [m]$, so for $(Bv)_i$ to be zero, all $kn$ independent trials need to miss index $i$. This happens for a single trial with probability $1 - 1/m$, and so the probability that all $kn$ trials miss $i$ is $(1 - 1/m)^{kn}$. It follows that the expected number of nonzeros

$$\mathbb{E}[\kappa] = \mathbb{E}[\|Bv\|_0] = \mathbb{E}[\sum_{i \in [m]} \|B_{i*}v\|_0] = \sum_{i \in [m]} \mathbb{E}[\|B_{i*}v\|_0] = m(1 - (1 - 1/m)^{kn}) = m(1 - p_n),$$

as claimed. To show (8), we use (9) and (10), which imply

$$1 - \frac{k\ell}{m-1} \le \exp(-k\ell/(m-1)) \le \left(1 - \frac{1}{m}\right)^{k\ell} = p_\ell \le \exp(-k\ell/m),$$

as claimed. The last claim is immediate. $\square$

**Theorem** (Restatement of Theorem 18). *Let $v, w \in \{0,1\}^d$ represent sets $X, Y$ respectively, and define the following counts:*

$$n \equiv |X \cap Y| = v \cdot w, \qquad n_v \equiv |X \setminus Y| = \|v\|_0 - n, \qquad n_w \equiv |Y \setminus X| = \|w\|_0 - n$$

*Let $x = 1 \wedge Bv$ and $y = 1 \wedge Bw$, where $B \in \{0,1\}^{m \times d}$ is a sparse binary matrix.*

*Assume WLOG $n_w \ge n_v$. Then for $\delta \in (0,1)$ and $\varepsilon > 0$, there are $k = O(\varepsilon^{-1} \log(1/\delta))$ and $m = O(k\varepsilon^{-1}(n_v n_w + n^2 + \varepsilon(n + n_w)))$ such that $h_{m,k}(x \cdot y) = n \pm \epsilon$ with failure probability $\delta$. Here, $h_{m,k}(z)$ (see Lemma 41) maps the Bloom filter output $1 \wedge Bz$ to an estimate of $\|z\|_0$.*

When $\varepsilon < 1/2$, the integrality of $n$ implies an exact result. Thus, the theorem implies that when $n$ and $n_v$ are $O(1)$, $m = O(kn_w)$ and $k = O(\log(1/\delta))$ suffice. This includes the case of membership testing, where $n_v = 1$.

The theorem also allows $\varepsilon > 1$, including $\varepsilon = \tilde{\varepsilon} n$, which allows us to estimate $n$ up to relative error $\varepsilon$ rather than additive error $\varepsilon$. In this case, $m = O(k\tilde{\varepsilon}^{-1}(n_v n_w/n + n) + kn_w)$ and $k = O(\max\{1, \tilde{\varepsilon}^{-1} n^{-1} \log(1/\delta)\})$ suffice.

*Proof.* Let $v^c = v \circ w$, where $\circ$ denotes element-wise product, so $n = \|v^c\|_0$, and let

$$\kappa_\cap = \|Bv^c\|_0.$$

Let $x^c = 1 \wedge Bv^c$, $\tilde{x} = x - x^c$, $\tilde{y} = y - x^c$, and let

$$\kappa_\Delta \equiv \tilde{x} \cdot \tilde{y} = x \cdot y - \kappa_\cap,$$

noting that $\tilde{x} \cdot x^c = 0$ and $\tilde{y} \cdot x^c = 0$,

$$
\begin{aligned}
\tilde{x} \cdot \tilde{y} &= x \cdot y - x^c \cdot x - x^c \cdot y + x^c \cdot x^c \\
&= x \cdot y - (x^c \cdot \tilde{x} + x^c \cdot x^c) - (x^c \cdot \tilde{y} + x^c \cdot x^c) + x^c \cdot x^c \\
&= x \cdot y - x^c \cdot x^c
\end{aligned}
$$

In other words, we can rewrite $x \cdot y = \kappa \equiv \kappa_\cap + \kappa_\Delta$, where $\kappa_\cap$ counts the 1s that $x$ and $y$ have in common due to elements in $\mathrm{supp}(v) \cap \mathrm{supp}(w)$, while $\kappa_\Delta$ counts the 1s common to $x, y$ due to elements corresponding to the symmetric difference $\mathrm{supp}(|v - w|) = \mathrm{supp}(v) \Delta \mathrm{supp}(w)$, that are not already counted in $\kappa_\cap$.

Recall that we defined $p_\ell \equiv (1 - 1/m)^{k\ell}$ in Lemma 41. Then,

$$
p_n \equiv 1 - \frac{1}{m}\mathbb{E}[\kappa_\cap] = 1 - \frac{1}{m}\mathbb{E}[\|Bv\|_0], \quad p_{n_v} \equiv 1 - \frac{1}{m}\mathbb{E}[\|Bv\|_0], \quad p_{n_w} \equiv 1 - \frac{1}{m}\mathbb{E}[\|Bw\|_0]
$$

Using Lemma 41 and (8), and letting $p_{vw} \equiv (1 - p_{n_v})(1 - p_{n_w})$,

$$
\begin{aligned}
\mathbb{E}[\kappa_\cap] &= m(1 - p_n) \\
\mathbb{E}[\kappa_\Delta] &= mp_n p_{vw}, \text{ and } p_{vw} \leq \frac{kn_v}{m-1} \cdot \frac{kn_w}{m-1} = \frac{k^2 n_v n_w}{m^2}(1 + O(1/m)).
\end{aligned}
\tag{12}
$$

where the latter is due to the fact that the probability that given entries $\tilde{x}_i$ and $\tilde{y}_i$ contribute to $\tilde{x} \cdot \tilde{y}$ is the probability that (a) the $n$ entries of $v^c$ yield $B_{i*}v^c = 0$, and (b) that both entries $\tilde{x}_i$ and $\tilde{y}_i$ are nonzero. Note that the three events in (a) and (b) are independent, since $\mathrm{supp}(v^c)$, $\mathrm{supp}(v - v^c)$, and $\mathrm{supp}(w - v^c)$ are disjoint.

Suppose, for some $\varepsilon_\cap, \varepsilon_T > 0$,

$$
\begin{aligned}
\kappa_\cap &\geq \mathbb{E}[\kappa_\cap] - \varepsilon_\cap \\
\kappa &\leq \mathbb{E}[\kappa] + \varepsilon_T,
\end{aligned}
\tag{13}
$$

which holds with small failure probability, as shown below. First, using these assumptions on $\kappa_\cap$ and $\kappa$, we establish a lower bound on $h_{m,k}(\kappa)$. Since $\kappa_\Delta \geq 0$,

$$
\begin{aligned}
-\frac{k}{\tilde{m}} h_{m,k}(\kappa) = \log(1 - \frac{\kappa}{m}) &= \log(1 - \frac{\kappa_\cap + \kappa_\Delta}{m}) \\
&\leq \log(1 - \frac{\kappa_\cap}{m}) \\
&\leq \log(1 - \frac{\mathbb{E}[\kappa_\cap] - \varepsilon_\cap}{m}) \\
&= \log(p_n + \frac{\varepsilon_\cap}{m}) \\
&= \log(p_n) + \log(1 + \frac{\varepsilon_\cap}{mp_n}) \\
&\leq \log(p_n) + \frac{\varepsilon_\cap}{mp_n} \\
&\leq -\frac{k}{\tilde{m}}(n - \frac{\varepsilon_\cap}{kp_n}), \text{ using (11) and the definition of } p_\ell, \text{ and so} \\
h_{m,k}(\kappa) &\geq n - \frac{\varepsilon_\cap}{kp_n}.
\end{aligned}
$$

If we assume that $m \geq 2kn + 1$, then $p_n \geq 1/2$. We have

$$
h_{m,k}(\kappa) - n \geq -\frac{\varepsilon_\cap}{kp_n} \geq -\frac{2\varepsilon_\cap}{k},
\tag{14}
$$

Again, using our assumptions on the sizes of $\kappa_\cap$ and $\kappa$, we can obtain an upper bound on $h_{m,k}(\kappa)$

$$
\begin{aligned}
h_{m,k}(\kappa) &= \frac{-\tilde{m}}{k} \log(1 - \frac{\kappa}{m}) \\
&\leq \frac{-\tilde{m}}{k} \log(1 - \frac{\mathbb{E}[\kappa] + \varepsilon_T}{m}) \\
&= \frac{-\tilde{m}}{k} \log(p_n - p_n p_{vw} - \frac{\varepsilon_T}{m}) \\
&= n - \frac{\tilde{m}}{k} \log(1 - p_{vw} - \frac{\varepsilon_T}{m p_n}) \\
&\leq n - \frac{m}{k} \log(1 - p_{vw} - \frac{\varepsilon_T}{m p_n}) \text{ using (11)} \\
&\leq n + \frac{m}{k} p_{vw} + \frac{m}{k} \frac{\varepsilon_T}{m p_n (1 - p_{vw})}, \text{ using (9)}
\end{aligned}
\tag{15}
$$

Assume that

$$
m \geq \max\{k\sqrt{2n_v n_w} + 1, 2kn + 1\}.
\tag{16}
$$

which includes the prior assumption on $m$. Then using (8) and (12),

$$
1 - p_{vw} \geq 1 - \frac{kn_v}{m-1} \cdot \frac{kn_w}{m-1} \geq 1/2
$$

so $p_n(1 - p_{vw}) \geq 1/4$. Therefore, from (15),

$$
\frac{k}{m}(h_{m,k}(\kappa) - n) \leq p_{vw} + \frac{\varepsilon_T}{m p_n (1 - p_{vw})} \leq p_{vw} + \frac{4\varepsilon_T}{m},
$$

and so

$$
h_{m,k}(\kappa) - n \leq \left[ \frac{kn_v n_w}{m} + \frac{4\varepsilon_T}{k} \right] (1 + O(1/m)), \text{ using(12)}.
\tag{17}
$$

Both (14) and (17) control how close $h_{m,k}(\kappa)$ is to $n$, but they rely on the assumptions in (13). We now need to bound $\Pr[\kappa_\cap - \mathbb{E}[\kappa_\cap] \leq -\varepsilon_\cap]$ and $\Pr[\kappa - \mathbb{E}[\kappa] \geq \varepsilon_T]$, so that (13) holds with high probability.

Recall that $\kappa_\cap = \|Bv^c\|_0$. We can express $\kappa_\cap$ as $\kappa_\cap(X_1, X_2, \ldots, X_{kn})$, where each $X_\ell$ makes a random choice $i \in [m]$ and sets $B_{i*}v^c$ to one; such choices are made $k$ times for each of the $n$ entries of $v^c$ that are equal to one. To apply Theorem 24, bounds on $B(\kappa_\cap)$ and the $\mathbf{Var}_i(\kappa_\cap)$ are needed.

Given choices for all other $X_{\ell'}$, a choice $i$ for $X_\ell$ changes $\kappa_\cap$ if and only if given those other choices, $B_{i*}v$ is equal to zero. The only effect of these other choices is the number of bins already with ones, that is, the number of $i'$ with $B_{i'*}v^c = 1$. Given that $r$ bins have ones, the expected value of $\kappa_\cap$ with respect to the random choice $X_\ell$ is $1 - r/m$, and the most $\kappa_\cap$ can differ from this is $1 - r/m \leq 1$, since it can hit one of those $r$ bins that are already one. The variance $\mathbf{Var}_\ell(\kappa_\cap)$ is $r/m - (r/m)^2 \leq r/m \leq (kn-1)/m$, the variance of a Bernoulli random variable with probability $1 - r/m$ of being one, and zero otherwise. Thus $\tilde{\sigma}(\kappa_\cap) \leq (kn)^2/m$.[4] From Theorem 24, if $\varepsilon_\cap = k\varepsilon/4$ for given $\varepsilon > 0$, then

$$
\begin{aligned}
\Pr[\kappa_\cap - \mathbb{E}[\kappa_\cap] \leq -\varepsilon_\cap] &\leq \exp\left(-\frac{2\varepsilon_\cap^2}{\tilde{\sigma}(\kappa_\cap) + \varepsilon_\cap B(\kappa_\cap)/3}\right) \leq \exp\left(-\frac{2\varepsilon_\cap^2}{(kn)^2/m + \varepsilon_\cap/3}\right) \\
&= \exp\left(-\frac{2(k\varepsilon/4)^2}{(kn)^2/m + (k\varepsilon/4)/3}\right) = \exp\left(-\frac{\varepsilon^2}{8n^2/m + 2\varepsilon/3k}\right).
\end{aligned}
\tag{18}
$$

Leaving aside for the moment the values needed for $m$ and $k$, we consider next $\Pr[\kappa - \mathbb{E}[\kappa] \geq \varepsilon_T]$. Here from $\kappa = \kappa_\cap + \kappa_\Delta$, we can regard $\kappa$ as the function of $2kn + 2kn_v + 2kn_w$ random variables, of which the first $2kn$, from $\kappa_\cap$, have $\tilde{\sigma}$ contribution at most $(kn)^2/m$ and $B \leq 1$. The next $2kn_v$,

---

[4]Put in a common setting, the bound of Broder & Mitzenmacher (2004) corresponds to putting into the tail estimate denominator a value of $kn \geq (kn)^2/m$ for $m \geq kn$, and the bound of Kamath et al. (1995) quoted in Bose et al. (2008) corresponds to $m \geq (kn)^2/m$ for $m \geq kn$.

representing the choices made that result in $\tilde{x}$, each have the property that their mean and variance, as a function of the choices of all the other variables, depend only on the number of ones in $\tilde{y}$: the minimum mean is $1 - kn_w$, so the contribution to $B$ is at most $1 - kn_w \leq 1$; the maximum variance is $(kn_w/m)(1 - (kn_w/m)) \leq kn_w/m$. There are $kn_v$ such variables, so the contribution to $\tilde{\sigma}$ is $(kn_v)(kn_w/m)$. Similarly, the last $kn_w$ random choices have $B \leq 1$ and contribution to $\tilde{\sigma}$ of $(kn_w)(kn_v/m)$. So $B(\kappa) \leq 1$, and $\tilde{\sigma}(\kappa) \leq k^2(n^2 + 2n_v n_w)/m$. If $\varepsilon_T = k\varepsilon/4$, we have

$$
\begin{aligned}
\Pr[\kappa - \mathbb{E}[\kappa] \geq \varepsilon_T] &\leq \exp\left(-\frac{2\varepsilon_T^2}{\tilde{\sigma}(\kappa) + \varepsilon_T B(\kappa)/3}\right) \leq \exp\left(-\frac{2\varepsilon_T^2}{k^2(n^2 + 2n_v n_w)/m + \varepsilon_T/3}\right) \\
&= \exp\left(-\frac{2(k\varepsilon/4)^2}{k^2(n^2 + 2n_v n_w)/m + (k\varepsilon/4)/3}\right) \\
&= \exp\left(-\frac{\varepsilon^2}{8(n^2 + 2n_v n_w)/m + 2\varepsilon/3k}\right).
\end{aligned}
\tag{19}
$$

If (19) yields a bound of $\delta/2$, (18) will too, and the total failure probability will be at most $\delta$.

Assume WLOG that $n_w \geq n_v$. From (17), for the mean of $\kappa$ to be at most $\varepsilon$, we must have $m \geq kn_v n_w/2\varepsilon$. This implies that in (19), $8(n^2 + 2n_v n_w)/m + 2\varepsilon/3k \leq 8n^2/m + c_1\varepsilon/k$, for $c_1 = 32 + \frac{2}{3}$. For $8n^2/m$ to not dominate $c_1\varepsilon/k$ in the sum $8n^2/m + c_1\varepsilon/k$, we need $m \geq 8kc_1 n^2/\varepsilon$, yielding $2c_1\varepsilon/k$ in the denominator, so that under these assumptions regarding $m$,

$$
\begin{aligned}
\Pr[\kappa_\Delta - \mathbb{E}[\kappa_\Delta] \geq \varepsilon] &\leq \exp\left(-\frac{\varepsilon^2}{8(n^2 + 2n_v n_w)/m + 2\varepsilon/3k}\right) \\
&\leq \exp\left(-\frac{\varepsilon^2}{2c_1\varepsilon/k}\right) = \exp(-k\varepsilon/2c_1) \leq \delta/2 \text{ for } k = 2c_1 \log(2/\delta)/\varepsilon.
\end{aligned}
$$

With a similar bound for $\Pr[\kappa - \mathbb{E}[\kappa] \leq -\varepsilon]$, via (14), we have that $m \geq \frac{k}{\varepsilon}(n_v n_w/2 + 8c_1 n^2 + \varepsilon(n + n_w))$ with $k = 2c_1 \log(2/\delta)/\varepsilon$ suffices to obtain $h_{m,k}(x \cdot y) = n \pm \varepsilon$ with failure probability $\delta$. The result follows. □

## E.2 COUNTING BLOOM FILTERS

In another sparse binary model, bundling is done by addition in instead of disjunction, so a bundle is simply $Bv$ instead of $1 \wedge Bv$. In the data structures literature, this is a *counting* variant of a Bloom filter. Here $B$ is chosen to be sparse in a slightly different way than for Bloom filters: each atomic vector $B_{*j}$ for $j \in [m]$ has all vectors with $k$ ones equally likely, or equivalently, is created by starting from a vector of all zeros, in each trial picking $i \in [m]$ uniformly at random, setting $B_{ij} \to 1$, until there are $k$ nonzeros.

When the sparsity is block-structured, that is, an atomic $m$-vector has $m$ divisible by $k$, and there are $k$ blocks each with exactly one nonzero entry, then the resulting bundling is called a *count-min sketch* in the data structures literature, and the Sparse Block Codes model in the VSA literature. See Cormode & Muthukrishnan (2005) and Laiho et al. (2015) respectively for more information.

Cormode & Muthukrishnan (2005) showed that dot product estimation for two bundles can be done as a nonlinear operation: taking dot products of corresponding blocks and then the minimum of these $k$ dot products. The resulting estimate $X$ of $v \cdot w$ using $Bv$ and $Bw$ has $v \cdot w \leq X \leq v \cdot w + \varepsilon$ with failure probability $\delta$, for $m = km'$ where $k = O(\log(1/\delta))$ and $m' = O(\|v\|_1 \|w\|_1/\varepsilon)$. For $v, w \in \{0,1\}^d$, the dimension $m$ needed for estimating set intersection size has a smaller dependence on $\varepsilon$, but possibly worse dependence on $\|v\|_1$, $\|w\|_1$, and $v \cdot w$ than what we showed for Bloom filters. This is because $\|v\|_1 \|w\|_1 = (n_v + n)(n_w + n) \geq n_v n_w + n^2$, where $n$, $n_v$, and $n_w$ are defined in Theorem 18.

Beyond the setting above, $v$ and $w$ may have entries that are nonnegative integers, corresponding to multi-sets (sets where each element appears with some multiplicity). Dot products for such multi-sets have applications in database operations. However, one could also consider an analog of set intersection for multi-sets where the multiplicity of an element in the intersection is the minimum of its multiplicities in the two intersected multi-sets. The size of the intersection, defined in this way, can also be estimated using bundles of sparse binary atomic vectors. This operation will be

analyzed next. For $x, y \in \mathbb{R}^m$, let $x \wedge y$ denote the vector $z$ with $z_i = \min\{x_i, y_i\}$. As discussed, on nonnegative vectors, we define the operation $x \wedge y \equiv \|x \wedge y\|_1 = \sum_{i \in [m]} \min\{x_i, y_i\}$. Note that $\|x - y\|_1 = \|x\|_1 + \|y\|_1 - 2(x \wedge y)$, analogous to $\|x - y\|_2^2 = \|x\|_2^2 + \|y\|_2^2 - 2(x \cdot y)$.

**Theorem** (Restatement of Theorem 19). *Let $v, w \in \mathbb{R}_{\geq 0}^d$. Let $K_b \equiv \|v - w\|_\infty \leq \max\{\|v\|_\infty, \|w\|_\infty\}$, and define:*

$$n \equiv v \wedge w, \qquad n_v \equiv \|v\|_1 - n, \qquad n_w \equiv \|w\|_1 - n,$$

*Let $x = Bv$ and $y = Bw$, where $B \in \{0,1\}^{m \times d}$ is a sparse binary matrix. Then, for $\delta \in (0,1)$ and $\varepsilon > 0$, there are $k = O(K_b \varepsilon^{-1} \log(1/\delta))$ and $m = 12\pi^2 k \varepsilon^{-1} n_v n_w = O(K_b \varepsilon^{-2} n_v n_w \log(1/\delta))$ such that*

$$(x \wedge y)/k - (v \wedge w) \in [0, \epsilon)$$

*with failure probability at most $\delta$. Since $n_v + n_w = \|v - w\|_1$, $m = O(K_b \varepsilon^{-2} \|v - w\|_1^2 \log(1/\delta))$ also suffices (using the AM-GM inequality on $n_v n_w$). If $\|v\|_1, \|w\|_1$ are stored, then $\|v - w\|_1$ can be estimated up to additive $\varepsilon$ with the same $m$.*

*Proof.* Let $\tilde{v} \equiv v - (v \wedge w)$, $\tilde{w} \equiv w - (v \wedge w)$ and $\tilde{x} \equiv B\tilde{v}$, $\tilde{y} \equiv B\tilde{w}$. Note that $n_v = \|\tilde{v}\|_1$, $n_w = \|\tilde{w}\|_1$. Using $\min\{c + a, c + b\} = c + \min\{a, b\}$, we have

$$\begin{aligned}
\frac{1}{k}(x \wedge y) &= \frac{1}{k}(Bv \wedge Bw) \\
&= \frac{1}{k}(B(v \wedge w) + B\tilde{v}) \wedge (B(v \wedge w) + B\tilde{w}) \\
&= \frac{1}{k}\vec{1}_m^\top B(v \wedge w) + \frac{1}{k}(B\tilde{v} \wedge B\tilde{w}) \\
&= v \wedge w + \frac{1}{k}(\tilde{x} \wedge \tilde{y}).
\end{aligned} \tag{20}$$

It remains to bound $\frac{1}{k}(\tilde{x} \wedge \tilde{y})$. We have $\tilde{v} \circ \tilde{w} = 0$, so for every $i \in [m]$, $B_{i*}\tilde{v}$ is independent of $B_{i*}\tilde{w}$. Therefore,

$$\Pr[\tilde{x}_i \wedge \tilde{y}_i \geq z] = \Pr[\tilde{x}_i \geq z] \Pr[\tilde{y}_i \geq z]$$

Since $\|\tilde{v}\|_1 = n_v$, $\mathbb{E}[\tilde{x}_i] = k n_v/m$, and so by Markov's inequality $\Pr[\tilde{x}_i \geq z] \leq k n_v/(zm)$. Similarly, $\Pr[\tilde{y}_i \geq z] \leq k n_w/(zm)$. Using the identity $\mathbb{E}[Y] = \sum_{z \geq 1} \Pr[Y \geq z]$,

$$\begin{aligned}
\mathbb{E}[\tilde{x}_i \wedge \tilde{y}_i] &= \sum_{z \geq 1} \Pr[\tilde{x}_i \wedge \tilde{y}_i \geq z] = \sum_{z \geq 1} \Pr[\tilde{x}_i \geq z] \Pr[\tilde{y}_i \geq z] \\
&\leq \sum_{z \geq 1} (k n_v/(mz))(k n_w/(mz)) = \zeta(2) \cdot \frac{k^2 n_v n_w}{m^2} = \frac{\pi^2}{6} \cdot \frac{k^2 n_v n_w}{m^2},
\end{aligned}$$

where $\zeta(2) = \sum_{z \geq 1} 1/z^2$ is the Riemann zeta function. Thus

$$\mathbb{E}[\tilde{x} \wedge \tilde{y}] \leq \frac{\pi^2}{6} \cdot \frac{k^2 n_v n_w}{m}. \tag{21}$$

It remains to show that $\tilde{x} \wedge \tilde{y}$ concentrates. We will use the Bernstein form of McDiarmid's inequality (Theorem 24), showing concentration of a function $f(X)$. Here $X \in [m]^{n_t}$ for $n_t \equiv k\|\tilde{v} + \tilde{w}\|_0 \leq k(n_v + n_w)$, and each $X_r$, for $r \in [n_t]$, maps to a column $j$ of $B$ where $\tilde{v}_j + \tilde{w}_j > 0$, and for random $i = X_r \in [m]$, $B_{ij}$ is incremented by one. The function $f(X)$ is then $f(X) = B\tilde{v} \wedge B\tilde{w}$, where $B$ is determined by $X$. To use Theorem 24, we need to bound $\tilde{\sigma}(f)$ and $B(f)$ of that theorem, bounding the effects of changes of single entries of $X$.

Suppose $B'$ is the result of the choices of $X$, for all entries except for entry $X_r$, where $X_r$ maps to column $j$, and to a random $I \in [m]$. Then the resulting $B = B' + e_I e_j^\top$, where $e_j \in \{0,1\}^d$ (with the 1 in the $j$ position) and $e_I \in \{0,1\}^m$ (with the 1 in the $I$ position) are natural basis vectors, and $Bv = B'v + e_I e_j^\top v = B'v + v_j e_I$.

Here $g_i(X, y)$ of Theorem 24 corresponds to $g(B', j, I) = (B' + e_I e_j^\top)\tilde{v} \wedge (B' + e_I e_j^\top)\tilde{w}$, and we need to bound, over all $B'$ and $j$, $\mathbf{Var}_i[g(B', j, i)]$ and $\max_i |g(B', j, i) - \mathbb{E}_{i'} g(B', j, i')]|$. Again

noting that because addition distributes over minimization, we can let $x' \equiv B'\tilde{v} - (B'\tilde{v} \wedge B'\tilde{w})$ and $y' \equiv B'\tilde{w} - (B'\tilde{v} \wedge B'\tilde{w})$, and write

$$
\begin{aligned}
g(B', j, I) &= (B'\tilde{v} + e_I \tilde{v}_j) \wedge (B'\tilde{w} + e_I \tilde{w}_j) \\
&= [(B'\tilde{v})_I + \tilde{v}_j] \wedge [(B'\tilde{w})_I - \tilde{w}_j] + \sum_{i \neq I} (B'\tilde{v} \wedge B'\tilde{w})_i \\
&= [(B'\tilde{v})_I + \tilde{v}_j - ((B'\tilde{v})_I \wedge (B'\tilde{w})_I)] \\
&\quad \wedge [(B'\tilde{w})_I + \tilde{w}_j - ((B'\tilde{v})_I \wedge (B'\tilde{w})_I)] \\
&\quad + ((B'\tilde{v})_I \wedge (B'\tilde{w})_I) + \sum_{i \neq I} (B'\tilde{v} \wedge B'\tilde{w})_i \\
&= (x'_I + \tilde{v}_j) \wedge (y'_I + \tilde{w}_j) + (B'\tilde{v} \wedge B'\tilde{w}) \\
&= (B'\tilde{v} \wedge B'\tilde{w}) + \beta_{Ij},
\end{aligned}
\tag{22}
$$

where $\beta_{Ij} \equiv (x'_I + \tilde{v}_j) \wedge (y'_I + \tilde{w}_j)$.

Since the choices of $I$ and $j$ don't affect $(B'\tilde{v} \wedge B'\tilde{w})$, only $\beta_{Ij}$ affects $\tilde{\sigma}(f)$ and $B(f)$. As $\tilde{v} \circ \tilde{w} = 0$, we first assume that $\tilde{v}_j > \tilde{w}_j = 0$. Under that assumption, since $x' \circ y' = 0$, either $\beta_{Ij} = 0$ or $\beta_{Ij} = y'_I \wedge \tilde{v}_j$, and so $\|\beta_{*j}\|_\infty \leq \|\tilde{v}\|_\infty$ for all $j$.

Allowing for the possibility that $\tilde{w}_j > \tilde{v}_j = 0$ and that sometimes $\beta_{ij} = 0$, we have

$$
\begin{aligned}
B(f) &= \max_{B', j, i} |g(B', j, i) - \mathbb{E}_{I'} g(B', j, I')]| \\
&= \max_{B', j, i} |\beta_{ij} - \mathbb{E}_{I'} \beta_{I'j}]| \qquad \text{from (22)} \\
&\leq \max\{\|\tilde{v}\|_\infty, \|\tilde{w}\|_\infty\} = \|v - w\|_\infty = K_b.
\end{aligned}
\tag{23}
$$

Also, under the assumption for given $j \in [d]$ that $\tilde{v}_j > \tilde{w}_j = 0$,

$$
\begin{aligned}
\mathbf{Var}_r(f) &= \sup_{B'} \mathbb{E}_I[(\beta_{Ij} - \mathbb{E}_{I'}\beta_{I'j}])^2] \leq \sup_{B'} \mathbb{E}_I[\beta_{Ij}^2] \leq \sup_{B'} \mathbb{E}_I[(y'_I \wedge \tilde{v}_j)^2] \text{ using } \tilde{v}_j > \tilde{w}_j = 0 \\
&\leq \sup_{B'} \tilde{v}_j \mathbb{E}_I[y'_I \wedge \tilde{v}_j] \leq \sup_{B'} \tilde{v}_j \mathbb{E}_I[y'_I] = \sup_{B'} \tilde{v}_j \|y'\|_1/m \\
&\leq k \tilde{v}_j \|\tilde{w}\|_1/m.
\end{aligned}
$$

Combining this with an analogous expression when $\tilde{w}_j > \tilde{v}_j = 0$, and summing over all $X_r$,

$$
\tilde{\sigma}(f) \leq 2k^2 \|\tilde{v}\|_1 \|\tilde{w}\|_1/m = 2k^2 n_v n_w/m.
\tag{24}
$$

We can now apply Theorem 24, scaling by $k$ as in the theorem, and using $k = \frac{2K_b}{3}\varepsilon^{-1}\log(1/\delta)$ and $m = 12\pi^2 k\varepsilon^{-1} n_v n_w = 8\pi^2 \varepsilon^{-2} n_v n_w \log(1/\delta)$, to obtain

$$
\begin{aligned}
\Pr[\tilde{x} \wedge \tilde{y} > \mathbb{E}[\tilde{x} \wedge \tilde{y}] + k\varepsilon/2] &\leq \exp\left(-\frac{2(k^2\varepsilon^2/4)}{\tilde{\sigma}(f) + k\varepsilon B(f)/6}\right) \\
&\leq \exp\left(-\frac{k^2\varepsilon^2/2}{(2k^2 n_v n_w/m) + k\varepsilon K_b/6}\right) \qquad \text{from (24), (23)} \\
&= \exp\left(-\frac{\varepsilon^2}{4n_v n_w/m + k^{-1}\varepsilon K_b/3}\right) = \exp\left(-\frac{\varepsilon^2}{n_v n_w/(3\pi^2\varepsilon^{-2} n_v n_w \log(1/\delta)) + \varepsilon^2/2\log(1/\delta)}\right) \\
&\leq \exp\left(-\frac{\varepsilon^2}{\varepsilon^2/2\log(1/\delta) + \varepsilon^2/2\log(1/\delta)}\right) = \delta.
\end{aligned}
$$

We also need (21) to bound

$$
\mathbb{E}[\tilde{x} \wedge \tilde{y}] \leq \frac{\pi^2}{6} \cdot \frac{k^2 n_v n_w}{m} = \frac{\pi^2}{6} \cdot \frac{k^2 n_v n_w}{12\pi^2 k\varepsilon^{-1} n_v n_w} = k\varepsilon/2.
$$

Putting these together,

$$
\Pr[\tilde{x} \wedge \tilde{y} \geq k\varepsilon] \leq \delta,
$$

which with (20) implies the result. $\qquad \square$

