# OpenReview forum: "Capacity Analysis of Vector Symbolic Architectures"
_ICLR.cc/2024/Conference — Submitted to ICLR 2024_

### Official Review · Reviewer_q9j4 · 2023-10-26

**Soundness:** 3 good
**Presentation:** 1 poor
**Contribution:** 2 fair
**Rating:** 5
**Confidence:** 3

**Summary:**

The paper contributes to the emerging field of hyper dimensional computing (HDC) (1) by considering a setting where high-dimensional vectors are first reduced to a lower-dimensional space and (2) by studying the limits in terms of dimensionality under which the original input properties among vectors produced by the HDC algebra are preserved in this “reduced space”. The paper provides a compilation of dimensionality bounds under different representation/operator settings (i.e. boolean/binary/sparse vectors, integer/binary bundling). From a high level perspective, the bounds are obtained by relying on the Johnson-Lindestrauss property.

**Strengths:**

1. Considerable effort has been invested in expanding the theory of HDC to the realm of "dimensionality reduction." The results obtained are not straightforward and, as far as I'm aware, appear to be original. I haven't had the chance to review the proofs, but the outcomes are reasonable. My assessment on the correctness should be therefore considered as an educated guess (**Originality/Quality**).

**Weaknesses:**

There are two major weaknesses which hinder the potential of this work:
1. While the analysis is remarkable, it is not directly obvious why the considered setting of dimensionality reduction is relevant. Indeed, HDC relies on the idea of the “blessing of dimensionality”, namely that by increasing the dimensionality nice properties like orthogonality and algebraic compositionality emerge. In a sense the proposed analysis focuses on the opposite direction, that is reducing the dimensionality. Please refer to the questions for more detailed discussion (**Significance**)
2. It is certainly outstanding the capability of distilling all the technical results in a 9 page paper. However, this comes at the cost of weak presentation quality (**Clarity**). Indeed, the paper appears more as a “list of bounds”. More discussion and explanation would be appreciated in order to valorise the obtained results. More concretely, it would be good to:
    1. Include a list of existing bounds on the value of dimensionality for the input space to have a fair comparison with the new derived ones and being able to better appreciate their significance.
    2. Provide some simple yet not trivial examples or simulations explaining or visualising the results of the different theorems
    3. Improve some sentences (see MINOR)

**MINOR**

For instance:

“Typically, the goal is to not only show the fixed point-property for some x …” -> Do you mean studying the stability of fixed-point solutions under random perturbations?

BSDC-CDT -> what does this stand for?

**Questions:**

Regarding significance:
1. Could you please explain why the dimensionality reduction aspect is relevant?
2. Could you please explicitly show case what is the improvement in terms of representation capacity with respect to existing bounds?
3. In Section 3.2, why the Hopfield network weights are parameterised by lower rank matrices? And why is this interesting? Similarly for the extended version named Hopfield+-.

Regarding clarity:
1. Can you please provide some simple yet non trivial examples explaining the main results from the different theorems? Alternatively, providing some simulations of the results contextualized to some practical problem?

I’m willing to increase my score if the above-mentioned weaknesses are resolved.

---

### Official Review · Reviewer_29HD · 2023-10-31

**Soundness:** 3 good
**Presentation:** 2 fair
**Contribution:** 2 fair
**Rating:** 3
**Confidence:** 2

**Summary:**

The Johnson-Lindenstrauss lemma is applied to study the properties of several vector-symbolic architectures (VSA) and memory
architectures (Bloom, Hopfield). The idea is that many VSA and memory tasks can be rewritten as the ability to perform a random projection from a specific representation, with bounded error derived from JL properties. Over 15 bounds are presented.

**Strengths:**

* The idea of applying the JL lemma to all those problems seems refreshing and useful.
 * The authors a proficient in applying the method to many different problems and could derive over 15 different bounds on the ability to perform various operations with bounded error.

**Weaknesses:**

Maybe it's a cultural thing in my subfield, but the manuscript does not meet my expectations with respect to a research paper. A new technique was applied to over 10 different settings and yielded interesting results. This should have been the (great) start of the manuscript, followed by comparing the results with known results for the specific problems.

 * In some cases, the results are known, so you would usually not publish them as a contribution (maybe a comment in the context of other results). I don't see known results in this manuscript; only really old references from the 1970s-1980s and recent reviews, but not specific results for comparison for the problem at hand.
 * In some cases, the results are not known but are not true, maybe due to a mistake or because of some nuanced assumption of the theorems. You need to provide proper simulations of the relevant problem to catch those.
 * In some cases, the results are not known and true but not interesting, maybe because they are vacuous bounds or because they are just worse than previously found ones. Here, a comparison with known results is in place.

I don't claim this manuscript should have contained all of this; this is too much for a single manuscript. If this research path is followed as suggested, it may give rise to 4-5 manuscripts (e.g., on VSA, on Bloom filters, on Hopfield).

I am not very familiar with VSA, but am very familiar with the Hopfield model, so let me focus on this:
 * There is a vast literature on storage capacity (with robust margin or without), and I can't tell if the new result is new or not and if it improves upon known results or not.
 * The capacity phase transition is simple to see in simulations, and I don't know if it resides with the result derived here.

**Questions:**

* What are the known results in the literature? Do any of the results improve them?
 * Have you conducted any simulations to see if there is a quantitative difference in the behaviour of the VSA (or Hopfield) below and above the bound?

---

### Official Review · Reviewer_tEi4 · 2023-11-20

**Soundness:** 2 fair
**Presentation:** 2 fair
**Contribution:** 1 poor
**Rating:** 3
**Confidence:** 2

**Summary:**

The representation capacity of a number of similar vector-based architectures are studied. Specifically, Vector Symbolic Architectures, Hopfield networks, and Bloom filters are studied. Over 15 bounds for these architectures are proven.

**Strengths:**

The analysis of bounds is rigorously carried out over an impressive number of different model configurations.

**Weaknesses:**

The major weakness of this article in the context of submission to ICLR is relevancy. The main contribution concerns a theoretical analysis of the bounds of representation capacity, which appears pertinent for ICLR. While there is little discussion on learning, theoretical analysis on the information capacity of a learnable representation could certainly fit within the scope of ICLR. However, there are two axes for concern with relevancy: the relevancy of VSAs, and the relevancy of the provided analysis.

The pertinence of VSAs to learning representations is described in the introduction relying on previous publications on VSA. Of those references, many are unpublished and a single author (D Kleyko) is cited twelve times, with some citations being simply for demonstration of application cases "biological times series data (Kleyko et al., 2018a; 2019b; Burrello et al., 2019)." There are two sentences in the current article relating VSAs to other, more common, forms of representation learning, notably deep learning, and the example provided is unclear (see Questions). From my understanding, VSAs are similar to very wide dense neural network layers, and are in some cases binarized. If this is the case, relating VSAs to existing work on high-dimensional neural network layers could lead to better understanding and greater relevancy of the approach. See, for example:

Jacot, A., Gabriel, F., & Hongler, C. (2018). Neural tangent kernel: Convergence and generalization in neural networks. Advances in neural information processing systems, 31.

Arora, S., Du, S. S., Hu, W., Li, Z., Salakhutdinov, R. R., & Wang, R. (2019). On exact computation with an infinitely wide neural net. Advances in neural information processing systems, 32.

The second relevancy concern is with the results of the analysis. This article exhaustively covers over 15 different bounds for VSAs, and pages 5 through 9 are almost exclusively used for the computation of these bounds. As such, the findings are never discussed or analyzed. Do the proven bounds differ from or build on existing understanding of this bound? Does the understanding of these bounds increase the utility of VSAs, or give insight into how their representations should be formed? Section 3.2 goes in the right direction in its discussion of Hopfield networks, however it stops short of analyzing the proven bounds. See, for example, the following works which study information capacity properties in order to improve or analyze representation learning:

Cheng, H., Lian, D., Gao, S., & Geng, Y. (2018). Evaluating capability of deep neural networks for image classification via information plane. In Proceedings of the European Conference on Computer Vision (ECCV) (pp. 168-182).

Gabrié, M., Manoel, A., Luneau, C., Macris, N., Krzakala, F., & Zdeborová, L. (2018). Entropy and mutual information in models of deep neural networks. Advances in Neural Information Processing Systems, 31.

Von Oswald, J., Niklasson, E., Randazzo, E., Sacramento, J., Mordvintsev, A., Zhmoginov, A., & Vladymyrov, M. (2023, July). Transformers learn in-context by gradient descent. In International Conference on Machine Learning (pp. 35151-35174). PMLR.

**Questions:**

Why is a VSA a good representation for neural network inputs, as mentioned in paragraph 3? The provided explanation is to "support inputs with different numbers of features," but neural networks are used in many cases where input dimensionality varies, for example via convolution or recurrence.

---

### Meta-Review · Area_Chair_2YMV · 2023-12-10

**Metareview:**

The paper describes an extensive analysis of the representational capacity of difference vector symbolic architectures. While the reviewers commonly acknowledge the effort in comparing the different architectures, they all question the significance and relevance of the results.

**Justification For Why Not Higher Score:**

The concerns with regard to significance and relevance are too large for an acceptable paper.

**Justification For Why Not Lower Score:**

N/A

---

### Decision · Program_Chairs · 2024-01-16

Reject